# Exploiting Contextual Objects and Relations for 3D Visual Grounding

**Li Yang**[1*], **Chunfeng Yuan**[1†*], **Ziqi Zhang**[1], **Zhongang Qi**[2], **Yan Xu**[3], **Wei Liu**[1], **Ying Shan**[2]
**Bing Li**[1], **Weiping Yang**[4], **Peng Li**[5,6], **Yan Wang**[5,6], **Weiming Hu**[1,7,8]

[1]State Key Laboratory of Multimodal Artificial Intelligence Systems (MAIS), CASIA
[2]ARC Lab, Tencent PCG, [3]The Chinese University of Hong Kong
[4]Education Management Information Center, Ministry of Education [5]Alibaba Group
[6]Zhejiang Linkheer Science And Technology Co., Ltd.
[7]School of Artificial Intelligence, University of Chinese Academy of Sciences
[8]School of Information Science and Technology, ShanghaiTech University

## Abstract

3D visual grounding, the task of identifying visual objects in 3D scenes based on natural language inputs, plays a critical role in enabling machines to understand and engage with the real-world environment. However, this task is challenging due to the necessity to capture 3D contextual information to distinguish target objects from complex 3D scenes. The absence of annotations for contextual objects and relations further exacerbates the difficulties. In this paper, we propose a novel model, CORE-3DVG, to address these challenges by explicitly learning about contextual objects and relations. Our method accomplishes 3D visual grounding via three sequential modular networks, including a text-guided object detection network, a relation matching network, and a target identification network. During training, we introduce a pseudo-label self-generation strategy and a weakly-supervised method to facilitate the learning of contextual objects and relations, respectively. The proposed techniques allow the networks to focus more effectively on referred objects within 3D scenes by understanding their context better. We validate our model on the challenging Nr3D, Sr3D, and ScanRefer datasets and demonstrate state-of-the-art performance. Our code will be public at `https://github.com/yangli18/CORE-3DVG`.

## 1 Introduction

Teaching machines to interpret and interact with the real world through vision and language has been a long-standing pursuit in artificial intelligence research. An important and fundamental task towards this goal is visual grounding, which involves detecting the referred visual object based on natural language input. While previous works have largely focused on visual grounding within 2D images [1, 2, 3, 4, 5], there has been a growing interest in extending this task to point clouds of 3D scenes, giving rise to 3D visual grounding [6, 7].

In 3D scenes, objects exhibit diverse spatial distributions, and it is common for objects of the same category to coexist. To clearly describe the target object, language descriptions often provide the contextual information, *i.e.* the contextual objects as well as their relations to the target object within the environment. For instance, in Figure 1(a), the text describes the target chair by specifying its relative position to the door and table. The interpretation of such contextual information from textual descriptions to visual representations is crucial for 3D visual grounding.

---

*Equal contribution, †Corresponding author

37th Conference on Neural Information Processing Systems (NeurIPS 2023).

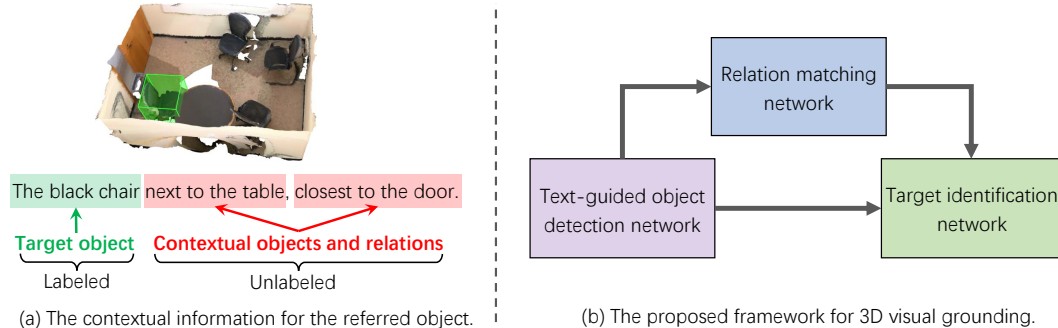

(a) The contextual information for the referred object.

(b) The proposed framework for 3D visual grounding.

Figure 1: (a) Language descriptions often provide the necessary contextual information to accurately depict the target objects. (b) Our proposed framework for 3D visual grounding.

Existing methods for 3D visual grounding typically involve matching a large set of object proposals with the textual description. These methods either use graph networks [6, 8, 9] or transformer architectures [10, 11, 12, 13, 14] to implicitly model contextual information between object proposals while performing cross-modal fusion with the text. Despite the effectiveness, their optimization objectives mainly focus on the final matching of object proposals with the textual description, without paying special attention to the alignment with the contextual information in the text. Consequently, important contextual cues may be overlooked during target inference, resulting in less reliable results.

In contrast, humans naturally possess a remarkable ability to infer the target object based on a language description. We can easily capture the relevant contextual information in the 3D scene to reason about the target indicated by the description in a two-step manner: Firstly, we locate all the objects mentioned in the text. Then we figure out the relations between these objects and compare them with the textual description, which enables us to ultimately identify the target referred to in the description. Motivated by this, we improve 3D visual grounding by mimicking this human reasoning process. Specifically, **(1)** we explicitly detect all mentioned contextual objects in addition to the target object, focusing the network on these objects to capture contextual information for target inference. As most current datasets do not have annotations for contextual objects, we propose a pseudo-label self-generation strategy, which leverages object semantics learned at the phrase level to generate learning labels for contextual objects. **(2)** We explicitly model the contextual relations between the candidate targets and the contextual objects as features of an adjacency matrix, which initially encode various spatial relations of objects. To facilitate the learning of contextual relations, we introduce a weakly-supervised learning loss that explicitly distinguishes related and unrelated relations for the target object.

The above ideas are embodied as a novel transformer-based framework that explicitly exploits **C**ontextual **O**bjects and **RE**lations for **3D** **V**isual **G**rounding (CORE-3DVG). Our framework, illustrated in Figure 1(b), comprises three modular networks that divide the 3D visual grounding process into three inference steps. Firstly, a text-guided object detection network performs joint detection of both the target and contextual objects. Next, a relation matching network models the representations of their contextual relations. Finally, the target identification network pinpoints the referred object based on the modeled contextual relations.

To summarize, our contributions are threefold:

- We propose to explicitly model and learn the objects and relations mentioned in the context to focus the network on contextual information for 3D visual grounding.

- We introduce a novel framework for 3D visual grounding, dubbed CORE-3DVG, which tackles the task of 3D visual grounding through three sequential modules. Our framework first detects all mentioned objects in the scene, then models their contextual relations, and finally performs context-based target inference.

- We evaluate our method on the challenging benchmarks including Nr3D, Sr3D, and Scan-Refer. Our method outperforms the previous state-of-the-art counterparts by significant margins. Extensive experiments validate the efficacy of our proposed techniques.

## 2 Related Work

**2D/3D Visual Grounding.** Visual grounding aims to locate target objects in 2D images or 3D point clouds based on the language input. Existing methods are generally categorized as one-stage methods [15, 16, 17, 4, 18] and two-stage methods [1, 19, 3, 20, 21, 22, 23, 24, 25, 26, 6, 7]. One-stage approaches fuse textual features with visual representations to generate target predictions in one shot. Two-stage methods follow a two-step detection and matching process: they first leverage the off-the-shelf detectors [27, 28, 29, 30] to generate object proposals, which are then matched with language input to select the best match as the estimated target.

The prevalent methods in 3D visual grounding adopt a two-stage scheme. Early works [6, 8, 9] use graph-based methods to model spatial relations among object proposals. With the rise of transformers [31], recent works [10, 12, 13, 14, 18] have adopted transformers for feature fusion and cross-modal fusion. Among them, SAT [12] incorporates 2D visual grounding to improve the performance of 3D visual grounding. Multi-view transformer [14] models different views in visual feature modeling.

**Context Modeling for Visual Grounding.** Modeling contextual information is important for visual grounding, especially for 3D visual grounding. While early works use graph neural networks or transformers to model contextual information, they fall short in encoding various spatial relations between objects. Recent 3DVG-transformer [11] and ViL3DRel [32] explicitly encode the objects' distances or orientations into their self-attention computation to better model 3D spatial relations. In contrast to prior works, our method focuses on the explicit learning of contextual objects and relations and exploiting this information for target inference.

## 3 Method

### 3.1 Framework

Our method focuses on the learning of contextual objects and relations, and exploits such information to achieve reliable 3D visual grounding. Figure 2 illustrates the overall framework. Given a 3D scene's point cloud and a language description, we initially encode their features into $F_v \in \mathbb{R}^{M \times C}$ and $F_l \in \mathbb{R}^{L \times C}$, respectively, where $M$ and $L$ represent the number of features and $C$ denotes the feature dimension. Based on these encoded features, 1) A text-guided object detection network first detects semantically related objects in the 3D scene, including target and contextual objects. 2) Then, we construct the spatial relation features in a pairwise manner for these detected objects and input them into the relation matching network for modeling contextual relations. 3) Finally, utilizing the modeled contextual features, the target identification network can effectively match the detected objects with the textual information to find the referred object.

#### 3.1.1 Text-Guided Object Detection Network

The object detection network, as illustrated in Figure 2(a), consists of $N_d$ transformer decoder layers. The network is initiated with a set of $K$ object queries $[o_1, o_2, \cdots, o_K]^{\mathrm{T}} \in \mathbb{R}^{K \times C}$, which are generated from the point cloud features $F_v$ based on their correlation with the textual features $F_l$ (see the supplementary material for more details). In each decoder layer, the initial object queries are first input to a self-attention layer to enable feature interaction between candidate objects. The queries are then passed through two cascaded cross-attention layers to gather textual and visual features successively. Next, a feed-forward network (FFN) updates the object queries. For the resulting $K$ object queries, two FFNs are applied to predict their object bounding boxes $[\hat{b}_1, \hat{b}_2, \cdots, \hat{b}_K]^{\mathrm{T}} \in \mathbb{R}^{K \times 6}$ and confidence scores $[\hat{p}_1, \hat{p}_2, \cdots, \hat{p}_K]^{\mathrm{T}} \in \mathbb{R}^{K \times 1}$, respectively.

To avoid duplicate object detections, we employ non-maximum suppression (NMS) and retain the top $T$ scoring detections. We represent the bounding boxes of the retained detections as $[\bar{b}_1, \bar{b}_2, \cdots, \bar{b}_T]^{\mathrm{T}} \in \mathbb{R}^{T \times 4}$ and their corresponding object queries as $[\bar{o}_1, \bar{o}_2, \cdots, \bar{o}_T]^{\mathrm{T}} \in \mathbb{R}^{T \times C}$. These detections correspond to the most relevant objects detected for the language description, including both target and contextual objects. For the training of this detection network, we propose a pseudo-label self-generation strategy to facilitate learning of unlabeled contextual objects, which is detailed in Section 3.2.

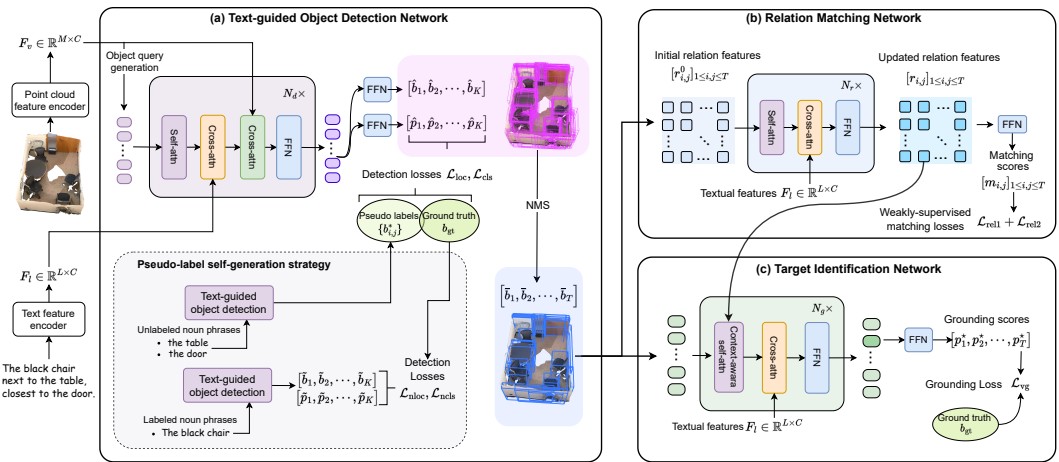

Figure 2: The overview of our framework. Our framework comprises three modular networks that divide the 3D visual grounding process into three inference steps.

### 3.1.2 Relation Matching Network

The relation matching network is developed to model the target object's contextual relations that match the textual description. Specifically, for each pair of detected objects $(\bar{b}_i, \bar{b}_j)$, we devise multiple spatial relation features that consider different aspects, including distance & orientation, volume & dimension, and perspective (see the supplementary material for details). This results in $T \times T$ spatial relation features $[r_{i,j}^s]_{1 \leq i,j \leq T}$ for the $T$ detected objects, where $r_{i,j}^s \in \mathbb{R}^{C_1}$ represents the relation feature with $\bar{b}_i$ as the candidate target and $\bar{b}_j$ as the contextual object. We perform linear projection on each feature $r_{i,j}^s$ and its corresponding contextual object query $\bar{o}_j$, and then concatenate them as the initial contextual relation feature:

$$r_{i,j}^0 = \text{Concat}(W_1 \cdot r_{i,j}^s, W_2 \cdot \bar{o}_j), \tag{1}$$

where $W_1 \in \mathbb{R}^{C/2 \times C_1}$ and $W_2 \in \mathbb{R}^{C/2 \times C}$ are the linear projection weights.

As depicted in Figure 2(b), the relation matching network takes the contextual relation features $[r_{i,j}^0]_{1 \leq i,j \leq T}$ as input and applies $N_r$ transformer decoder layers to model contextual relations that match the textual description. To achieve this, each decoder layer contains a self-attention layer for intra-modal information interaction among relation features, a cross-attention layer on textual features for cross-modal fusion, and an FFN to transform the relation features. The relation matching network outputs the updated relation features $[r_{i,j}]_{1 \leq i,j \leq T}$, and we apply a two-layer FFN to generate their respective matching scores w.r.t. the textual information, denoted as $[m_{i,j}]_{1 \leq i,j \leq T}$. A weakly-supervised learning approach is proposed to supervise these matching scores and thereby learn the contextual relations associated with the referred object (detailed in Section 3.3).

### 3.1.3 Target Identification Network

We propose the target identification network to find the referred object with fine-grained alignment to the textual description by leveraging the information of the detected objects and their contextual relations. As shown in Figure 2(c), with the detected objects as candidates, we input their object queries $[\bar{o}_1, \bar{o}_2, \cdots, \bar{o}_T]^{\text{T}} \in \mathbb{R}^{T \times C}$ into the target identification network. This network aligns the object queries with the textual features via $N_g$ transformer decoder layers. Each decoder layer includes a self-attention layer and a cross-attention layer to enable intra-modal and cross-modal feature fusion, followed by an FFN layer to refine the object query representations. We take the output object queries from the last decoder layer and apply a two-layer FFN to estimate their confidence scores as the target object, denoted as $[p_1^\star, p_2^\star, \cdots, p_T^\star]^{\text{T}} \in \mathbb{R}^{T \times 1}$.

To exploit contextual relations for target discrimination, we propose a context-aware self-attention mechanism incorporating relation features to better model contextual information among candidate objects. Specifically, the self-attention layer contains multiple attention heads to process the input object queries. In each attention head, the object queries are first linearly projected as the query, key,

and value embeddings, represented by $Q, K, V \in \mathbb{R}^{T \times d_k}$, respectively ($d_k$ is the dimension of the embeddings). The conventional self-attention method computes the attention of the $i$-th object to the $j$-th object by correlating their query and key embeddings, *i.e.* $\mathrm{softmax}_j(Q(i)K(j)^{\mathrm{T}}/\sqrt{d_k})$. Our self-attention mechanism extends this attention computation by considering the modeled contextual relations between objects. To accomplish this, we calculate an additional query embedding of the object queries as $Q_r \in \mathbb{R}^{T \times d_k}$, and perform linear projection on the contextual relation features $[r_{i,j}]_{1 \leq i,j \leq T}$ to obtain their key and value embeddings as $R_k, R_v \in \mathbb{R}^{T \times T \times d_k}$, respectively. The attention of the $i$-th object to the $j$-th object is then computed as follows:

$$\mathrm{attn}_{i,j} = \mathrm{softmax}_j\Big(\frac{Q(i)K(j)^{\mathrm{T}} + Q_r(i)R_k(i,j)^{\mathrm{T}}}{\sqrt{2 \cdot d_k}}\Big). \tag{2}$$

The resulting self-attention matrix allows each object to focus more on the objects that have important contextual relations between them. Moreover, we aggregate the value embeddings of both the object queries and their associated relation features for attention-based feature fusion:

$$\mathrm{out}_i = \sum_j \mathrm{attn}_{i,j} \cdot (V(j) + R_v(i,j)). \tag{3}$$

In this manner, each attention head of the self-attention layer achieves context-based feature aggregation, and the outputs from all attention heads are concatenated to update the input object queries. Consequently, the updated object queries can explicitly encode information about related contextual objects and relations. We incorporate the context-aware self-attention layer in each decoder layer of the target identification network, enabling a more effective comparison of modeled object queries with textual information to discern the referred object.

### 3.2 Pseudo-Label Self-Generation for Contextual Object Learning

Our method starts with a text-guided detection network to detect all the objects mentioned in the textual description. However, since the contextual objects associated with the referred targets are typically unlabeled, the object detection network lacks the requisite supervision for effective training. To address this issue, we propose a pseudo-label self-generation strategy that facilitates the learning of contextual objects within the network.

As shown in Figure 2(a), we first extract the noun phrases for physical objects from the input text and divide them into the *labeled noun phrase* (of the target object) and the *unlabeled noun phrases* (of the other contextual objects). We input these extracted noun phrases into the object detection network along with the same initial object queries and point cloud features. For the labeled noun phrase, the network produces $K$ detected bounding boxes $[\tilde{b}_1, \tilde{b}_2, \cdots \tilde{b}_K]^{\mathrm{T}} \in \mathbb{R}^{K \times 6}$ and confidence scores $[\tilde{p}_1, \tilde{p}_2, \cdots, \tilde{p}_K]^{\mathrm{T}} \in \mathbb{R}^{K \times 1}$. Given the ground-truth bounding box of the target object as $b_{\mathrm{gt}} \in \mathbb{R}^6$, we find the best matching detection result $\{\tilde{b}_\eta, \tilde{p}_\eta\}$ (with index $\eta$) as the positive sample. Then we define the following losses to supervise the detection results of the labeled noun phrase:

$$\begin{aligned}
\mathcal{L}_{\mathrm{nloc}} &= \mathcal{L}_{\mathrm{box}}(b_{\mathrm{gt}}, \tilde{b}_\eta), \\
\mathcal{L}_{\mathrm{ncls}} &= -\log(\tilde{p}_\eta) - \sum_{k \neq \eta} \lambda_{\mathrm{n}} \cdot \log(1 - \tilde{p}_k),
\end{aligned} \tag{4}$$

where the loss function $\mathcal{L}_{\mathrm{box}}$ consists of the GIoU loss [33] and the L1 loss, *i.e.* $\mathcal{L}_{\mathrm{box}}(\cdot, \cdot) = \lambda_{\mathrm{giou}}\mathcal{L}_{\mathrm{giou}}(\cdot, \cdot) + \lambda_{\mathrm{L1}}\mathcal{L}_{\mathrm{L1}}(\cdot, \cdot)$ ($\lambda_{\mathrm{giou}}$ and $\lambda_{\mathrm{L1}}$ are hyper-parameters to balance the two losses).

During training, based on the supervised learning for labeled noun phrases, we develop the network's ability to detect objects for various noun phrases. We thereby apply the detection network to the unlabeled noun phrases and utilize the output detection results to create pseudo labels for contextual objects. In particular, for each of the $U$ unlabeled noun phrases, we process the detection results with NMS and select the $H$ detections with the highest scores. We use $b_{i,j}^* \in \mathbb{R}^6$ and $p_{i,j}^* \in \mathbb{R}$ to denote the bounding box and unnormalized score of the $j$-th detection for the $i$-th unlabeled noun phrase. This gives $U \times H$ object detections $\{(b_{i,j}^*, p_{i,j}^*)\}_{1 \leq i \leq U, 1 \leq j \leq H}$ to serve as the pseudo labels for contextual objects. As shown in Figure 2(a), we combine these pseudo labels with the target's ground-truth label $b_{\mathrm{gt}}$ to supervise the network's detection results $\{(\hat{b}_i, \hat{p}_i)\}$ for the entire sentence (see Section 3.1.1). Similar to DETR [34], we use the Hungarian algorithm to find the best matching detections for the combined labels, where $\tau$ and $\sigma_{i,j}$ denote the matching indexes for the target label

and each pseudo label, respectively. Then the supervision losses on the detection results are computed as:

$$\mathcal{L}_{\text{loc}} = \mathcal{L}_{\text{box}}(b_{\text{gt}}, \hat{b}_\tau) + \sum_{i=1}^{U}\sum_{j=1}^{H} w_{i,j} \cdot \mathcal{L}_{\text{box}}(b_{i,j}^*, \hat{b}_{\sigma_{i,j}}),$$

$$\mathcal{L}_{\text{cls}} = -\log(\hat{p}_\tau) - \sum_{i=1}^{U}\sum_{j=1}^{H} w_{i,j} \cdot \log(\hat{p}_{\sigma_{i,j}}) - \sum_{\substack{k \neq \tau \\ k \notin \{\sigma_{i,j}\}}} \lambda_n \cdot \log(1 - \hat{p}_k),$$

(5)

where $w_{i,j} = \frac{\exp(p_{i,j}^*)}{\sum_{j=1}^{H} \exp(p_{i,j}^*)} \cdot \text{sigmoid}(p_{i,j}^*)$ denotes the learning weight for each pseudo label. We use $w_{i,j}$ to encourage learning pseudo labels of higher scores under each unlabeled noun phrase while also considering the individual confidence of each pseudo label to mitigate the impact of learning unreliable pseudo labels. With Equation (5), the network learns to understand the object phrases within the input text for associated object detection. During inference, the network detects target and contextual objects based on the input text without the need to manually extract their noun phrases.

### 3.3 Weakly-Supervised Contextual Relation Learning

Our relation matching network outputs a score matrix $[m_{i,j}]_{1 \leq i,j \leq T}$ for all pairs of $T$ detected objects, representing the matching scores between their contextual relations and the textual description. However, since there are no explicit labels for contextual relation learning, we propose a weakly-supervised method to find the contextual relations associated with the referred object. The overall optimization objectives for contextual relation learning are:

1. For the target object, there exists at least one contextual relation with a high matching score.

2. For the related contextual object, its contextual relations should have lower matching scores when paired with objects that are not the target.

To accomplish this, we first match the $T$ detected objects with the ground truth target $b_{\text{gt}}$ and find the best matching detection as $\bar{b}_\alpha$ (indexed by $\alpha$), which the model shall later infer as the detected target. Then we take the $\alpha$-th row from the score matrix $[m_{i,j}]_{1 \leq i,j \leq T}$, obtaining $[m_{\alpha,j}]_{j \neq \alpha}$ as the contextual relation matching scores between the detected target $\bar{b}_\alpha$ and other objects. From these scores, we find the maximum score $m_{\alpha,\beta}$ with column index $\beta = \arg\max_j([m_{\alpha,j}]_{j \neq \alpha})$. We thereby consider the object $\bar{b}_\beta$ as the most related contextual object for the target $\bar{b}_\alpha$. Following the optimization objectives 1 and 2, we devise the following losses to supervise the matching scores of the $\alpha$-th row and $\beta$-th column:

$$\mathcal{L}_{\text{rel1}} = \sum_{\substack{j=1 \\ j \neq \alpha}}^{T} \max(0, \ \Delta + m_{\alpha,j} - m_{\alpha,\beta}),$$

$$\mathcal{L}_{\text{rel2}} = \sum_{i=1}^{T} \max(0, \ \Delta + m_{i,\beta} - m_{\alpha,\beta}),$$

(6)

where the hyper-parameter $\Delta$ defines the score margin. The losses $\mathcal{L}_{\text{rel1}}$ and $\mathcal{L}_{\text{rel2}}$ ensure that the contextual relation $r_{\alpha,\beta}$ (between objects $\bar{b}_\alpha$ and $\bar{b}_\beta$) has a matching score surpassing the scores of other contextual relations (with target $\bar{b}_\alpha$ or contextual object $\bar{b}_\beta$) by a margin of $\Delta$. This encourages the network to explicitly differentiate related and unrelated contextual relations for the target object. We utilize the learned contextual relation features for target inference, as detailed in Section 3.1.3.

### 3.4 Training

In addition to the previous losses for learning contextual objects and relations, we also supervise the target identification network on its estimated grounding scores $[p_1^\star, p_2^\star, \cdots, p_T^\star]$ for the $T$ candidate object detections. We take the $\alpha$-th detected object that best matches the ground-truth label $b_{\text{gt}}$ as the positive sample, and we treat other detections as negative samples. We compute the binary

cross-entropy loss on their grounding scores as follows:

$$\mathcal{L}_{\text{vg}} = -\log(p_\alpha^\star) + \sum_{\substack{k=1 \\ k \neq \alpha}}^{T} \lambda_n \cdot \log(1 - p_k^\star). \quad (7)$$

We take the sum of losses for all the modular networks to train the entire model. Following the previous works [6, 7], we apply auxiliary losses to the predictions from every decoder layer of the modular networks. The overall training loss is:

$$\mathcal{L} = \frac{1}{N_d} \sum_{k=1}^{N_d} \left( \mathcal{L}_{\text{nloc}}^k + \mathcal{L}_{\text{ncls}}^k + \mathcal{L}_{\text{loc}}^k + \mathcal{L}_{\text{cls}}^k \right) + \frac{1}{N_r} \sum_{k=1}^{N_r} \left( \mathcal{L}_{\text{rel1}}^k + \mathcal{L}_{\text{rel2}}^k \right) + \frac{1}{N_g} \sum_{k=1}^{N_g} \mathcal{L}_{\text{vg}}^k, \quad (8)$$

where the superscript $k$ indexes the losses of predictions from multiple decoder layers.

## 4 Experiments

### 4.1 Datasets

**Nr3D** [6] is built on 3D indoor scene dataset ScanNet [35]. It contains 41,503 human-annotated text descriptions, covering 76 object categories and 707 indoor scenes. The dataset is divided into "Easy" and "Hard" subsets depending on whether there are objects that share the same category as the target in the scene. Based on whether a specific viewpoint is required to infer the target, the dataset is divided into "View-dep." and "View-indep." subsets.

**Sr3D** [6] contains 83,572 descriptions that are automatically generated using specific templates. Similar to the Nr3D dataset, the Sr3D dataset is divided into multiple subsets for evaluation.

**ScanRefer** [7] provides 51,583 text descriptions of 11,046 objects in 800 3D scenes from the ScanNet. On average, there are 13.81 objects and 64.48 text descriptions per scene. The official division takes 36,665 samples as the training set and 9,508 as the test set. According to whether the target object category is unique in the scene, the dataset is divided into "Unique" and "Multiple" subsets.

### 4.2 Implementation Details

We utilize the AdamW optimizer [36] to train our model with a batch size of 24. For visual feature encoding, we utilize the PointNet++ network [37] with an initial learning rate of $10^{-3}$. The rest of the model has an initial learning rate of $10^{-4}$, and the weight decay value is set to $5 \times 10^{-4}$. We employ the first three layers of the RoBERTa [38] to extract text features. We train our model for 120 epochs on the Nr3D dataset, 60 epochs on the Sr3D dataset, and 100 epochs on the ScanRefer dataset. The hyper-parameter $\Delta$ in Equation 6 is set to 0.4, and we set the weight hyper-parameters $\lambda_{\text{giou}} = 1$ and $\lambda_{\text{L1}} = 5$ for the $\mathcal{L}_{\text{box}}$ loss. The numbers of decoder layers for the text-guided object detection, relation matching, and target identification networks are set to $N_d = 3$, $N_r = 2$, and $N_g = 3$, respectively. Following the previous work [39], we apply rotation data augmentation for the 3D point cloud scenes and augment supervision with detection prompts during training. We measure accuracy using Acc@0.25 for both Nr3D and Sr3D datasets, where a predicted bounding box is considered correct if its IoU with the ground truth target exceeds 0.25. For the ScanRefer dataset [7], we employ two metrics: Acc@0.25 and Acc@0.5.

### 4.3 Comparison with State-of-the-Art Methods

Table 1 presents the comparative results of our method with the current state-of-the-art methods on the Nr3D, Sr3D, and ScanRefer datasets. Our method consistently outperforms the previous methods. For the Nr3D and Sr3D datasets, it is standard practice to use ground-truth object boxes as object proposals to infer the referred object. However, considering that ground-truth boxes are usually unavailable in practical scenarios, our model is designed to locate objects of interest using a dedicated detection network. Thus, for a fair comparison, we quote the evaluation results from BUTD-DETR [39], where previous methods are evaluated with detected object proposals by a pretrained 3D object detector [40]. The quoted results are marked with † and we denote this evaluation setup as "det" in Table 1. On the Nr3D and Sr3D datasets, as shown in Table 1, our method exhibits

Table 1: Comparison with the state-of-the-art methods on Nr3D, Sr3D, and ScanRefer.

| Method | Nr3D Acc@0.25 (det) | Sr3D Acc@0.25 (det) | ScanRefer Acc@0.25 | ScanRefer Acc@0.5 |
|---|---|---|---|---|
| ReferIt3DNet [6] | 24.0[†] | 27.7[†] | 26.4 | 16.9 |
| ScanRefer [7] | - | - | 35.5 | 22.4 |
| TGNN [8] | - | - | 37.4 | 29.7 |
| InstanceRefer [9] | 29.9[†] | 31.5[†] | 40.2 | 32.9 |
| FFL-3DOG [41] | - | - | 41.3 | 34.0 |
| LanguageRefer [13] | 28.6[†] | 39.5[†] | - | - |
| 3DVG-Transformer [11] | - | - | 45.9 | 34.5 |
| SAT 2D [12] | 31.7[†] | 35.4[†] | 44.5 | 30.1 |
| BUTD-DETR [39] | 43.3 | 52.1 | 52.2 | 39.8 |
| CORE-3DVG | **49.57** | **54.30** | **56.77** | **43.84** |

Table 2: Comparison with the state-of-the-art methods on different subsets of ScanRefer.

| Method | Unique Acc@0.25 | Unique Acc@0.5 | Multiple Acc@0.25 | Multiple Acc@0.5 | Overall Acc@0.25 | Overall Acc@0.5 |
|---|---|---|---|---|---|---|
| ReferIt3DNet[6] | 53.8 | 37.5 | 21.0 | 12.8 | 26.4 | 16.9 |
| ScanRefer[7] | 63.0 | 40.0 | 28.9 | 18.2 | 35.5 | 22.4 |
| TGNN[8] | 68.6 | 56.8 | 29.8 | 23.2 | 37.4 | 29.7 |
| InstanceRefer[9] | 77.5 | 66.8 | 31.3 | 24.8 | 40.2 | 32.9 |
| FFL-3DOG[41] | 78.8 | **67.9** | 35.2 | 25.7 | 41.3 | 34.0 |
| 3DVG-Transformer[11] | 77.2 | 58.5 | 38.4 | 28.7 | 45.9 | 34.5 |
| SAT 2D[12] | - | - | - | - | 44.5 | 30.1 |
| BUTD-DETR[39] | 84.2 | 66.3 | 46.6 | 35.1 | 52.2 | 39.8 |
| CORE-3DVG | **84.99** | 67.09 | **51.82** | **39.76** | **56.77** | **43.84** |

significant performance advantages over the previous methods. Compared with the recent BUTD-DETR, which employs a larger transformer encoder and decoder network, our method achieves absolute improvements of 6.27% and 2.20% on the Nr3D and Sr3D datasets, respectively.

Table 2 provides a detailed performance comparison between our method and previous approaches on the ScanRefer dataset. Our method shows superior performance across various test settings and evaluation metrics. Notably, on the challenging "Multiple" subset, where multiple objects share the same category as the target objects, our method achieves 51.82% in Acc@0.25, surpassing the leading BUTD-DETR [39] by an appreciable margin 5.22%. Our explicit modeling of contextual information enables better handling of such challenging scenarios. For the simpler "Unique" subset, where the target object is unique, we also achieve higher or comparable results compared to previous methods. Our overall accuracy on the ScanRefer dataset reaches 56.77% (Acc@0.25) and 43.84% (Acc@0.5), respectively, outperforming all the previous methods.

### 4.4 Ablation Studies

In this section, we conduct the ablation studies on the Nr3D [6] dataset to validate our method.

**Learning Contextual Objects and Relations.** Table 3 presents the ablation study on the proposed explicit learning of contextual objects and relations. The first row of Table 3 represents the baseline model, which achieves an overall accuracy of 34.66% (Acc@0.25). Based on this baseline, we first introduce the learning of contextual objects, resulting in a significant improvement of 9.76% in overall accuracy. The improvements are consistently observed across all test subsets, as shown in the second row of Table 3. While the contextual relations are not modeled, the contextual objects alone provide crucial contextual information for target inference. In the third row of the table, we separately introduce the learning of contextual relations and observe an improvement of 2.67%. The "View-dep" subset exhibits the largest improvement of 6.14%, highlighting the effectiveness of relation learning for context modeling. Finally, by incorporating the learning of both contextual objects and relations, we achieve an overall accuracy of 49.57%. This surpasses the baseline model by 14.91% and achieves the best performance among these ablation variants, as shown in the last row of Table 3.

Table 3: The ablation study of learning contextual objects and relations.

| Learning context. objects | Learning context. relations | Easy | Hard | View-dep. | View-indep. | Overall |
|---|---|---|---|---|---|---|
| | | 40.50 | 29.04 | 32.13 | 35.59 | 34.66 |
| ✓ | | 49.85 | 39.20 | 40.75 | 45.78 | 44.42 |
| | ✓ | 43.34 | 31.55 | 38.27 | 36.98 | 37.33 |
| ✓ | ✓ | 53.91 | 45.39 | 48.19 | 50.07 | **49.57** |

Table 4: The ablation study of pseudo-label self-generation strategy.

| Method | Easy | Hard | View-dep. | View-indep. | Overall |
|---|---|---|---|---|---|
| − | 40.50 | 29.04 | 32.13 | 35.59 | 34.66 |
| + Supervised learning on labeled noun phrases | 45.05 | 31.87 | 33.17 | 40.23 | 38.33 |
| + Learning pseudo labels for unlabeled noun phrases | 49.85 | 39.20 | 40.75 | 45.78 | **44.42** |

Table 5: The ablation study of initial spatial relation features.

| Distance & Orientation | Volume & Dimension | Perspective-related relations | Easy | Hard | View-dep. | View-indep. | Overall |
|---|---|---|---|---|---|---|---|
| | | | 49.85 | 39.20 | 40.75 | 45.78 | 44.42 |
| ✓ | | | 52.38 | 40.99 | 43.53 | 47.70 | 46.57 |
| | ✓ | | 51.62 | 40.70 | 42.94 | 47.20 | 46.05 |
| | | ✓ | 52.49 | 42.58 | 46.46 | 47.81 | 47.44 |
| ✓ | ✓ | ✓ | 52.55 | 43.42 | 46.85 | 48.28 | **47.90** |

**Pseudo-Label Self-Generation Strategy for Contextual Object Learning.** In Table 4, we investigate the pseudo-label generation strategy used for contextual object learning. First, we only employ the detection loss defined in Equation 4 to learn the objects corresponding to labeled noun phrases. The second row of Table 4 shows an overall accuracy improvement of 3.67% compared to the baseline model in the first row. This suggests that augmenting the textual diversity for object detection training also contributes to improved visual grounding. After that, we utilize the pseudo labels generated for unlabeled noun phrases to guide the learning of contextual objects. The third row of Table 4 demonstrates that this technique boosts the model's accuracy from 38.33% to 44.42% (+ 6.09%) and exhibits significant performance enhancements across all test subsets. These results further emphasize the importance of learning contextual objects, as it allows the model to effectively capture contextual information for target inference.

**Initial Spatial Relation Features.** We develop a diverse set of spatial relations between objects as initial features for modeling contextual relations. Table 5 presents an ablation study of these relation features. The first row of Table 5 shows an overall accuracy of 44.42% when the initial spatial relation features are not used. Then, we individually incorporate the relation features related to "Distance & Orientation", "Volume & Dimension", and "Perspective-related relations", improving the overall accuracy by 2.15%, 1.63%, and 3.35%, respectively. Notably, the perspective-related relation feature achieves the greatest improvement, especially in the "View-dep." test subset (from 40.75% to 46.46%), highlighting its effectiveness in reasoning textual descriptions associated with specific viewpoints. Finally, combining all these relation features in the last row of Table 5 yields the highest accuracy of 47.90%.

**Weakly-Supervised Contextual Relation Learning.** In Table 6, we conduct an ablation study on the learning of contextual relations. The first two rows of Table 6 show an overall accuracy improvement of 3.48% when directly using the initial spatial relation features. However, as shown in the third row of Table 6, the relation matching network alone fails to improve performance without the inclusion of the weakly-supervised losses in Equation 6. By incorporating weakly-supervised learning of contextual relations, we achieve an overall accuracy improvement of 1.77% and observe appreciable performance advancements across all test subsets, as depicted in the last row of Table 6. These results clearly demonstrate the necessity of the weakly-supervised learning in effectively modeling contextual relations.

Table 6: The ablation study of contextual relation learning.

| Initial spatial relation features | Relation matching network | Weakly-supervised learning losses | Easy | Hard | View-dep. | View-indep. | Overall |
|:---:|:---:|:---:|:---:|:---:|:---:|:---:|:---:|
| | | | 49.85 | 39.20 | 40.75 | 45.78 | 44.42 |
| ✓ | | | 52.55 | 43.42 | 46.85 | 48.28 | 47.90 |
| ✓ | ✓ | | 52.55 | 43.24 | 46.90 | 48.13 | 47.80 |
| ✓ | ✓ | ✓ | 53.91 | 45.39 | 48.19 | 50.07 | **49.57** |

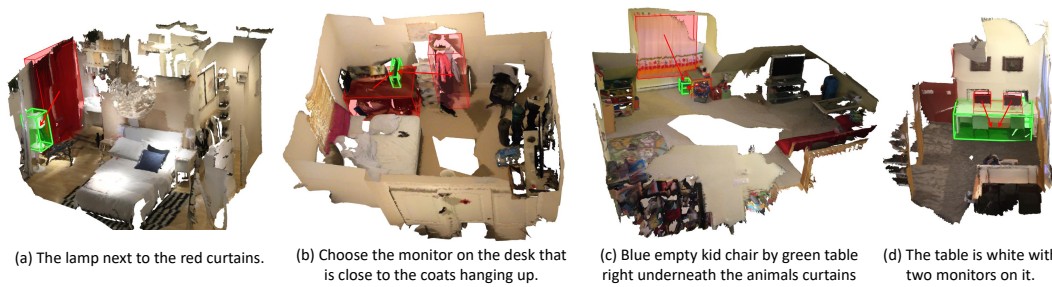

(a) The lamp next to the red curtains.

(b) Choose the monitor on the desk that is close to the coats hanging up.

(c) Blue empty kid chair by green table right underneath the animals curtains

(d) The table is white with two monitors on it.

Figure 3: 3D visual grounding results with associated contextual objects and relations.

## 4.5 Visualization

In Figure 3, we visualize the visual grounding results of our method for various descriptions and 3D scenes. We use green bounding boxes to represent the detected target objects and red bounding boxes to indicate the contextual objects that exhibit relation matching scores greater than 0.4. The results illustrate the efficacy of our method in comprehending contextual information within the text to infer the referred target object. For example, in Figure 3(a), when given the description "The lamp next to the red curtains", our model successfully locates the correct lamp and the adjacent red curtains, thereby avoiding misclassification of the lamp near the bed. This explicit utilization of contextual information enables a more reliable inference of the target object.

## 5 Conclusion

In this paper, we presented a 3D visual grounding framework that focuses on learning contextual objects and relations. The proposed framework consists of three sequential modules: text-guided object detection, relation matching, and target identification networks. The innovative techniques of pseudo-label self-generation and weakly-supervised learning facilitate the learning of contextual objects and relations. We verify the effectiveness of contextual objects and relations learning through extensive experiments and achieve leading performances on multiple benchmarks, including Nr3D, Sr3D, and ScanRefer.

Despite the remarkable performance achieved, the generalization ability of our method is still limited since our model has been only trained on datasets with limited object semantics and corpus. The performance of our algorithms could be influenced by the types of devices used to capture the point clouds and the types of 3D scenes. In the future, we plan to further dig into the generalization ability of 3D visual grounding frameworks and extend our method to more practical and realistic scenarios.

**Acknowledgements.** This work is supported by the National Key R&D Program of China (No. 2022ZD0118501), the Beijing Natural Science Foundation (JQ21017, L223003, M22005, 4224091), the Natural Science Foundation of China (Grant No. 61972397, 62222206, 62036011, 62192782, 61721004, U2033210, 62372451, 62192785, 62372082, 62202469), the Major Projects of Guangdong Education Department for Foundation Research and Applied Research (2018KZDXM066), Guangdong Provincial University Innovation Team Project (2020KCXTD045).

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
