# Appendix

## A  Object Query Generation

The text-guided object detection network, as described in Section 3.1.1, first takes $K$ object queries as input for detecting objects in the 3D scene. Here, we present the details of the process for generating object queries.

Detecting objects in a 3D scene using a fixed set of 3D anchor boxes or parameterized representations poses a challenge due to the large search space. Thus, we initially select a subset of features from the point cloud features $F_v \in \mathbb{R}^{M \times C}$ to serve as initial object queries for the subsequent detection process. Since the network focuses only on detecting objects that are semantically related to the text, we filter the point cloud features based on their semantic correlation with textual features. Specifically, based on a multi-head attention module [1], we take the point cloud features $F_v \in \mathbb{R}^{M \times C}$ as queries and the textual features $F_l \in \mathbb{R}^{L \times C}$ as keys and values. In this way, each point cloud feature captures the most relevant semantic information from the text, yielding the corresponding textual semantic features $F_s = \text{MHA}(F_v, F_l, F_l) \in \mathbb{R}^{M \times C}$. Subsequently, the point cloud features $F_v$ and the extracted semantic features $F_s$ are separately transformed using a two-layer MLP, resulting in $F_v'$ and $F_s'$. We compute the inner product between each point cloud feature and its corresponding semantic feature vector, followed by normalization using the sigmoid function to derive the relevance score $s_{corr}(i)$ $(1 \leq i \leq M)$:

$$s_{corr}(i) = \frac{1}{1 + \exp\left(-\langle F_v'(i), F_s'(i) \rangle\right)} \tag{1}$$

Finally, we directly select the top $K$ point cloud features with the highest correlation scores as the initial object queries $[o_1, o_2, \cdots, o_K]^{\text{T}} \in \mathbb{R}^{K \times C}$ for subsequent object detection.

## B  Spatial Relation Features

As mentioned in Section 3.1.2, for each pair of detected objects $(\bar{b}_i, \bar{b}_j)$, we devise multiple spatial relation features that consider different aspects, including distance & orientation, volume & dimension, and perspective. The computation of these spatial relation features is explained in detail below.

### B.1  Distance & Orientation

For each pair of objects $(\bar{b}_i, \bar{b}_j)$, we compute the differences in the coordinates of their bounding box centers on each axis, represented as $(\mathrm{d}x_{i,j}, \mathrm{d}y_{i,j}, \mathrm{d}z_{i,j})$. We also calculate the Euclidean distance $\mathrm{D}_{i,j}$ and $\mathrm{D}_{i,j}^{\text{xy}}$ to measure the spatial separation between their centers in both 3D space and the X-Y plane. Moreover, we consider the generalized IoU (GIoU) [2] between the two object bounding boxes, denoted as $\text{GIoU}(\bar{b}_i, \bar{b}_j)$, as a measure of their spatial distance relation.

The orientation between two objects is represented by encoding the angle values of the line that connects their centers in the spherical coordinate system. Specifically, for the line connecting the centers of objects $\bar{b}_j$ and $\bar{b}_i$, we denote its angle with the Z axis as $\theta_{i,j}$ and the angle of its projection line in the X-Y plane with the X axis as $\phi_{i,j}$. The sine and cosine of these angles are computed to depict the relative orientation between the two objects:

$$\begin{aligned}
\sin(\theta_{i,j}) = \mathrm{D}_{i,j}^{\text{xy}}/\mathrm{D}_{i,j}, \quad \cos(\theta_{i,j}) = \mathrm{d}z_{i,j}/\mathrm{D}_{i,j}, \\
\sin(\phi_{i,j}) = \mathrm{d}y_{i,j}/\mathrm{D}_{i,j}^{\text{xy}}, \quad \cos(\phi_{i,j}) = \mathrm{d}x_{i,j}/\mathrm{D}_{i,j}^{\text{xy}},
\end{aligned} \tag{2}$$

The above calculation results are combined as the spatial relation features of "Distance & Orientation".

### B.2  Volume & Dimension

Additionally, we encode the size relationship between object pairs. We denote the dimensions (length, width, height) and volume of the object bounding box $\bar{b}_i$ as $(l_i, w_i, h_i)$ and $v_i$, respectively. The dimensions and volume differences between object $\bar{b}_i$ and object $\bar{b}_j$ are calculated and then normalized by the sum of their respective dimensions and volume. The derived features representing the relations

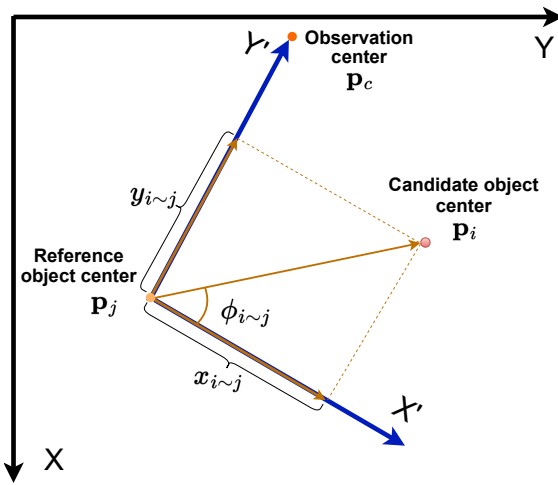

Figure 1: Perspective-related relation features.

in dimensions and volume between the objects are denoted as $f_{i,j}^s = [\mathrm{dl}_{i,j}, \mathrm{dw}_{i,j}, \mathrm{dh}_{i,j}, \mathrm{dv}_{i,j}]^{\mathrm{T}}$:

$$
\begin{aligned}
\mathrm{dl}_{i,j} &= \big(\mathrm{l}_i - \mathrm{l}_j\big)\big/\big(\mathrm{l}_i + \mathrm{l}_j\big), \quad \mathrm{dw}_{i,j} = \big(\mathrm{w}_i - \mathrm{w}_j\big)\big/\big(\mathrm{w}_i + \mathrm{w}_j\big), \\
\mathrm{dh}_{i,j} &= \big(\mathrm{h}_i - \mathrm{h}_j\big)\big/\big(\mathrm{h}_i + \mathrm{h}_j\big), \quad \mathrm{dv}_{i,j} = \big(\mathrm{v}_i - \mathrm{v}_j\big)\big/\big(\mathrm{v}_i + \mathrm{v}_j\big),
\end{aligned}
\tag{3}
$$

### B.3 Perspective-Related Relations

Many referring descriptions often specify a viewpoint relative to a reference object before describing the target's relative location. For this situation, we further construct perspective-related relation features to more accurately model the contextual relations of the target object. Perspective-related descriptions are usually influenced by the observation angle relative to the horizontal plane. For instance, two objects viewed from the front and back have opposite left-right positional relationships, so we construct perspective-related features on the X-Y plane.

Specifically, we first calculate the scene center coordinates $\mathbf{p}_c = [x_c, y_c]^{\mathrm{T}}$ from the point cloud data of the 3D scene, which serve as the starting position for observing different objects. As shown in Figure 1, for the target object $\bar{b}_i$ and reference object $\bar{b}_j$, their centers' coordinates on the X-Y plane are $\mathbf{p}_i = [x_i, y_i]^{\mathrm{T}}$ and $\mathbf{p}_j = [x_j, y_j]^{\mathrm{T}}$ respectively. Using $\mathbf{p}_j$ as the coordinate origin, we establish a new coordinate reference system using the direction from this point to the observation center $\mathbf{p}_c = [x_c, y_c]^{\mathrm{T}}$ as the Y′ axis. The unit vector in the Y′ axis direction is $\mathbf{u}_{\mathrm{Y}'} = (\mathbf{p}_c - \mathbf{p}_j)/\|\mathbf{p}_c - \mathbf{p}_j\|_2$, and a 90-degree rotation yields the unit vector $\mathbf{u}_{\mathrm{X}'}$ on the X′ axis, perpendicular to the Y′ axis. We compute the position vector of the center of object $\bar{b}_i$ relative to the center of object $\bar{b}_j$ and project it onto the X′ and Y′ axes by calculating its inner product with the two unit vectors. This yields the coordinates of the center of object $\bar{b}_i$ in the reference system of new perspective:

$$
\begin{cases}
x_{i\sim j} = \langle \mathbf{p}_i - \mathbf{p}_j, \mathbf{u}_{\mathrm{X}'} \rangle, \\
y_{i\sim j} = \langle \mathbf{p}_i - \mathbf{p}_j, \mathbf{u}_{\mathrm{Y}'} \rangle,
\end{cases}
\tag{4}
$$

Moreover, we calculate the sine and cosine of the angle $\phi_{i\sim j}$ between the line from the center of object $\bar{b}_j$ to the center of object $\bar{b}_i$ and the X′ axis:

$$
\begin{cases}
\sin(\phi_{i\sim j}) = y_{i\sim j}/\sqrt{(x_{i\sim j})^2 + (y_{i\sim j})^2}, \\
\cos(\phi_{i\sim j}) = x_{i\sim j}/\sqrt{(x_{i\sim j})^2 + (y_{i\sim j})^2}.
\end{cases}
\tag{5}
$$

Thus, utilizing this coordinate system based on the viewpoint direction from the scene center to the reference object $\bar{b}_j$, we derive new position and angle information for the object $\bar{b}_i$, denoted as $f_{i,j}^v = [x_{i\sim j}, y_{i\sim j}, \sin(\phi_{i\sim j}), \cos(\phi_{i\sim j})]^{\mathrm{T}}$. This feature represents the spatial relation associated with the aforementioned viewpoint direction.

Finally, we combine the above features related to distance & orientation, volume & dimension, and perspective-related relations. These combined features serve as the initial spatial relation features for contextual relation learning.

## C  More Visualization Results

Figure 2 illustrates our visual grounding results on some test samples with complex contextual descriptions. These descriptions contain information about multiple contextual objects and relations to specify the target objects. Despite not being specifically designed for modeling multiple contextual relations, our method can still detect all mentioned contextual objects and their relations. For example, in Figure 2(a), given the description "Choose the monitor on the desk that is close to the coats hanging up," the model not only detects the target monitor but also identifies the desk it is on and the coats hanging nearby. This enables finer information alignment during target inference, thus improving the reliability of the visual grounding results. The model also exhibits a good understanding of negative relation descriptions. For instance, in Figure 2, for the description "The only pillow that isn't on the bed," our model successfully detects not only the bed but also the other pillows on the bed, leveraging them as contextual information to determine the pillow that is not on the bed. Additionally, Figure 3 demonstrates our visual grounding results for small target objects. Our method effectively exploits contextual information to locate small targets.

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

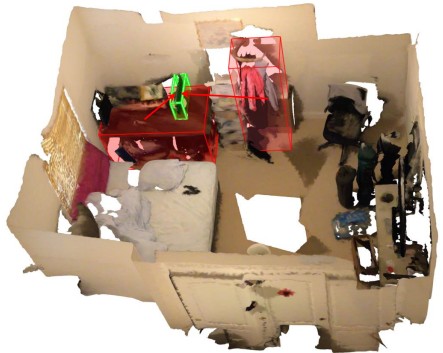

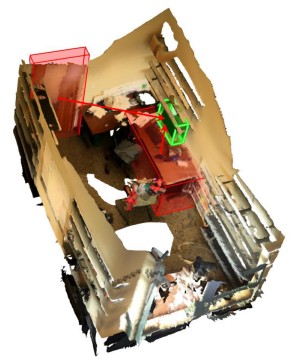

(a) Choose the monitor on the desk that is close to the coats hanging up.

(b) Choose the single monitor on the desk by the door.

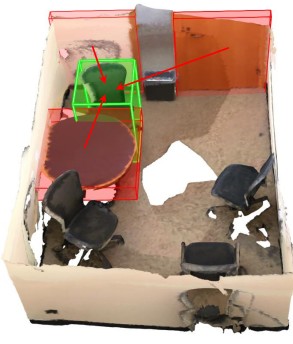

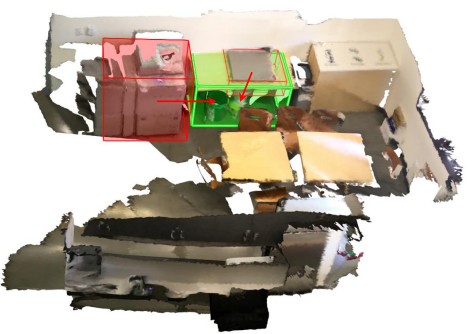

(c) The black chair next to the table , closest to the door.

(d) The table to the right of the copier printer with the green cutting board on it.

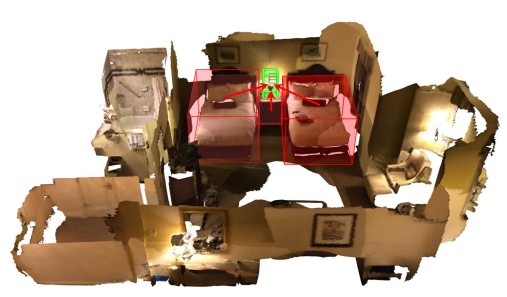

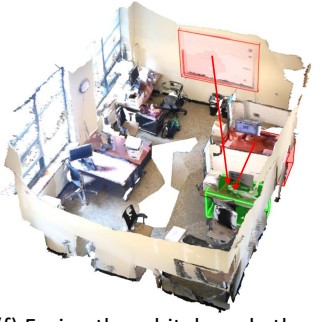

(e) The lamp between the beds. The lamp on the nightstand.

(f) Facing the whiteboard , the second desk from the right.

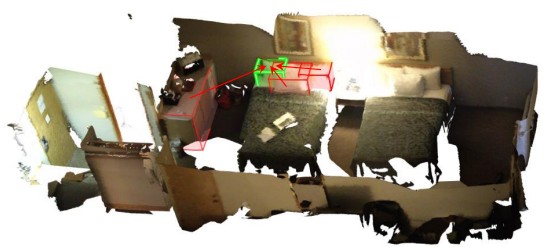

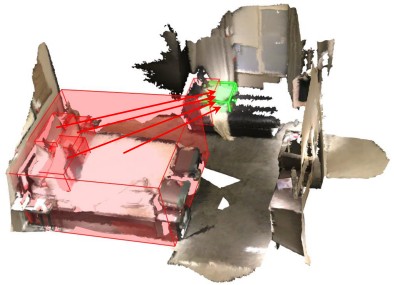

(g) The rear, left hand side pillow on the bed that is closest to the cabinets.

(h) The only pillow that isn't on the bed.

Figure 2: 3D visual grounding results for more complex contextual descriptions.

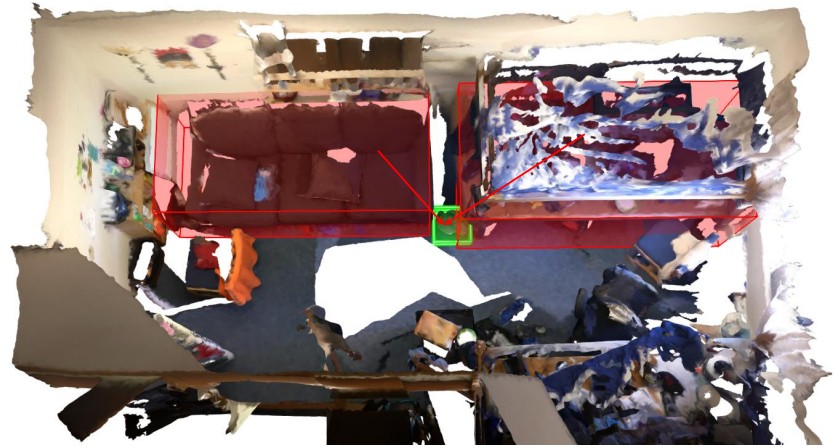

(a) Shoes are between couch and bed.

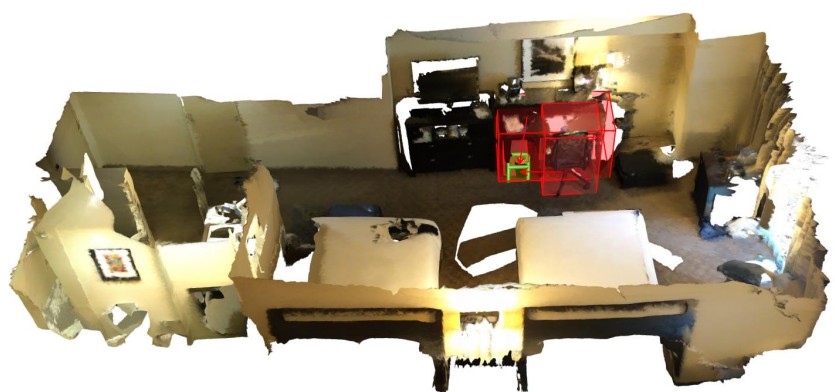

(b )The trash can is under the desk to the left of the chair.

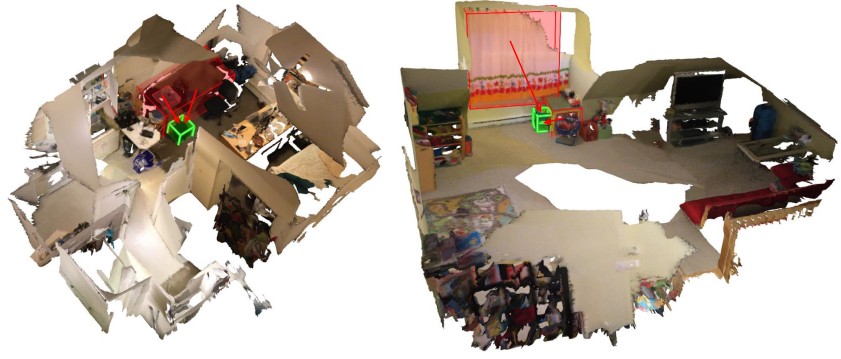

(c) Black backpack beneath the couch.

(d) Blue empty kid chair by green table right underneath the animals curtains

Figure 3: 3D visual grounding results for small target objects.