# OpenReview forum: "Exploiting Contextual Objects and Relations for 3D Visual Grounding"
_NeurIPS.cc/2023/Conference — NeurIPS 2023 poster_

### Official Review · Reviewer_VN72 · 2023-07-02

**Soundness:** 3 good
**Presentation:** 2 fair
**Contribution:** 3 good
**Rating:** 6
**Confidence:** 4

**Summary:**

The paper proposes to supervise the detection of contextual objects and contextual relations in a given language sentence to help improve language grounding performance. Specifically, the authors devise three modules: a) Language guided detection module is responsible for detecting all mentioned objects. Since the supervision for contextual objects is not readily available, this paper proposes a pseudo-label generation strategy to supervise them. b) Relation Matching Network which is responsible for enhancing relations between the target and contextual objects, with supervision derived from another pseudo labelling strategy proposed in the paper. c) Target Inference Network which is responsible for utilizing the detection queries from module (a) and relationship matrix from (b) to produce the final grounding output. Experiments show superior performance over SOTA baselines with significant margins. Ablations validate the claims made in the paper.

**Strengths:**

- SOTA performance by a significant margin
- Nice and thorough ablations, clearly showing what matters and what doesn’t

**Weaknesses:**

I read this paper multiple times and yet I do not understand it clearly. I think the paper writing can be improved to aid understanding. Below are some of my confusions:
- About Text-guided detection module:
    - My understanding is that the authors want to supervise the model with detection boxes for both target and contextual objects. Grounding datasets do not label contextual objects and hence the authors propose to use pseudo-label generation for the contextual objects. For that, they train the model to detect “noun phrases”. For eg: “The chair next to the table, closest to the door”, the model will be given a text utterance of “the chair” and will be asked to predict boxes of “the chair”.
        - Q1: do you use chair boxes of all chairs in the scene or the ground truth “chair” mentioned by the utterance “the chair next to ….. door”. If latter, wouldn’t it induce ambiguity in training because the model will be penalized for predicting extra chairs?
        - Q2: Instead of using pseudo labels, why not use the ground truth of the noun classes for the contextual objects as well. For example, in the above utterance, supervise the model with “all tables”, and “all doors” for contextual objects. This supervision is readily available in all referential datasets including Referit3D/ScanRefer via Scannet annotations.  Why pseudo-labels are better since anyway the model is supervised with predictions from “the table” supplied to detection model which will, in best case, produce all tables already available in detection setup. Is there any quantitative comparison for this?
        - Q3: How is the training exactly performed? Do you train first for some epochs to make the model train on noun phrases and then start using pseudo labels? And to get the pseudo labels, for each forward pass, you would need to run several passes through the model to get pseudo labels for each noun phrase?
        - Also, I think the idea of training on detection phrases/noun chunks is proposed by GLIP [1] and BUTD-DETR [34]. The idea of improving performance by training on intermediate/contextual objects is proposed in MDETR and BUTD-DETR [34]. It might be good to explicitly state that.  I think the novelty here is how to get the labels for intermediate objects for datasets like NR3D/ScanRefer where such a supervision does not exist.
	[1]: Li, Liunian Harold, et al. "Grounded language-image pre-training. 2022 IEEE." CVF Conference on Computer Vision and Pattern Recognition (CVPR). 2021.
[2]: Kamath, Aishwarya, et al. "Mdetr-modulated detection for end-to-end multi-modal understanding." Proceedings of the IEEE/CVF International Conference on Computer Vision. 2021.

<start> Minor Comments
- About Relation matching Network:
    - I think Section-3.3 could use a figure to make it more clear to understand.
- Target Inference Network:
    - The name “inference” is slightly misleading as it gives an impression that this is only used at test time and is not involved in training
    - I think Figure-2 does not indicate  any connection going from section-b to section-c i.e. how (and if) relation matching network’s output is used in target inference network.
- Overall, a pseudo code on how everything connects could aid the clarity tremendously I think.

<end> Minor Comments

- The dataset on which ablations are conducted is missing (which can be inferred to be NR3D from the numbers, but would be nicer to mention). Additionally, in my initial readings, I was confused why the final row in Ablation Table-4 gets 44.42% instead of the reported NR3D number of 49.57 (which later I realized is because Table-4 is concerned with ablation the jump from 34.66 -> 44.42 in Table-3).

- The related works section is not thorough and well-written. Some suggestions should be to discuss works related to supervising contextual objects and enhancing focus on contextual relations from both 2D and 3D community; identifying gaps in the literature and properly motivating how the proposed work is different.

- Minor Writing/Formatting errors:
    - Implementation details section is duplicated in main paper as well as supplementary
    - “References” heading duplicated

**Questions:**

My main questions are regarding the details of the proposed architecture as mentioned in the limitations section. I think addressing how the authors could improve the clarity of the paper would be the most important for improving my ratings of the paper. The proposed model performs very well and I think the results would be valuable to the community, hence I want to recommend acceptance. However, the paper  lacks clarity is very hard to read which can be significantly improved.

**Limitations:**

Yes

---

> ### Author Rebuttal · Authors · 2023-08-10
>
> We greatly appreciate your comprehensive feedback on our paper. Your questions and suggestions have been carefully addressed to improve the clarity and quality of the paper. Please find our responses below.
>
> **About Text-guided detection module:**
>
> **A1:**
> We supervise the model with the ground truth box of the "chair" mentioned by the utterance "the chair next to… door". We understand the concern regarding the model being penalized for predicting extra chairs. In our experiments, we empirically found that even with this supervision, the model is still capable of detecting multiple chairs in the scene, without overfitting to predict only one chair. This could be due to the difficulty of training the model to exactly match the object phrase "the chair" with just one labeled chair.  Additionally, we employ a smaller loss weighting hyper-parameter to alleviate this penalty during training.
>
> **A2:**
> Given the free-form language descriptions, the extracted noun phrases are not strictly the same as the standard noun classes. The extracted noun phrases might use different words to describe the object category, and they can encompass descriptive terms, including attributes and other modifiers.  As a result, it is not straightforward to associate labeled objects of specific categories with the noun phrase extracted from language descriptions. In our method, the extracted noun phrases of target objects naturally form a diverse set of object phrases with corresponding labeled objects. We train the network using this data to detect objects for various object phrases, which can be directly used to generate pseudo-labels for other objects mentioned in language descriptions.
>
> **A3:**
> Sorry for the ambiguities here. We do not devise a complex training process, and we start to use  pseudo labels from the beginning of model training. Considering the pseudo labels may not be initially accurate, we have devised a learning weight based on the score of each pseudo label (as shown in Eq. 5), which mitigates the impact of learning unreliable pseudo labels. To obtain the pseudo labels, we form a batch of extracted noun phrases and input them into the text-guided detection network for an extra forward pass. We will revise the related part to improve clarity.
>
> **A4:**
> We acknowledge the contributions of GLIP [1] and BUTD-DETR [34] in training on detection phrases/noun chunks, as well as the idea of training on intermediate/contextual objects as presented in MDETR and BUTD-DETR [34]. In the revised paper, we will explicitly mention these works. The novelty of our approach, as you rightly pointed out, involves the methodology we employ to obtain pseudo labels for intermediate objects. This is particularly important in datasets like NR3D/ScanRefer where such explicit supervision is absent. We will make this distinction more explicit in the revised manuscript to give due credit to prior works and clearly highlight our unique contributions.
>
> **Minor and Other Comments**
>
> - **Relation Matching Network:** We will incorporate a new figure in Section 3.3 to visually represent the processes involved, making it more intuitive.
> - **Target Inference Network:**
> Thanks for your advice and we will rename the module to avoid confusion. We will update Figure-2 to show how the output of the relation matching network is fed into the target inference network, to offer a clearer understanding of the flow.
> - We will add a pseudo code in Section 3.4 to provide a clearer overview of how the components are connected and their interactions.
> - In the experiment section, we will explicitly mention the dataset used for conducting the ablation study, and provide clearer illustrations of the experimental settings for the tables.
> - Thanks for your suggestions. We will improve the related work section as suggested. This should provide readers with a clearer picture of where our work fits into the existing literature.
> - Thank you for pointing these out. We will fix these errors.  Additionally, we will thoroughly review the paper for any other formatting or typographical errors.
>
> **Questions**
>
> Thank you for your constructive feedback and for recognizing the potential value of our work to the community. We acknowledge the concerns raised about the clarity of the paper, especially regarding the architecture details. We will revise the paper to provide a clearer description of the architecture and improve overall readability. We appreciate your recommendation and will ensure that the necessary revisions are made to enhance the paper's clarity.

---

> > ### Comment · Reviewer_VN72 · 2023-08-16
> >
> > Thank you for your reply and explanations to my questions. I think I have a better understanding of the architecture now.
> >
> > I see your point now that the noun phrase in the given utterance may not have a direct text-to-text matching with a ground truth object and that's why you chose to devise a pseudo-label strategy instead of using the available ground truths.
> >
> > My main concern is with regard to the paper's readability, but I think the proposed changes by the authors are good. I will vote to accept this paper, subject to discussion with other reviewers and AC.

---

> > > ### Author Response · Authors · 2023-08-20
> > >
> > > Dear Reviewer,
> > >
> > > Thank you for your comprehensive feedback and for taking the time to delve into the details of our work. We are pleased to hear that our explanations have provided clarity on the architecture and the motivations behind our choices.
> > >
> > > Your feedback has been instrumental in identifying areas where we can improve the paper's clarity, and we are committed to making the necessary revisions to enhance the paper's readability.
> > >
> > > Thank you again for your recommendation to accept the paper. We will endeavor to address all the concerns and if it is possible, we hope our efforts can further improve your final rating on this paper. 🤗
> > >
> > > Best regards!

---

### Official Review · Reviewer_oYfR · 2023-07-03

**Soundness:** 3 good
**Presentation:** 3 good
**Contribution:** 3 good
**Rating:** 6
**Confidence:** 5

**Summary:**

This paper presents a transformer-based method for learning underlying spatial relationships in scenes for 3D visual grounding.
Specifically, three neural modules are proposed to tackle this challenge: 1) a language-guided object detector; 2) a relation matching network; 3) an attention-based target inference module.
To facilitate learning the contextual relationships, this work introduces a pseudo-label generation strategy and a weakly supervised training objective during training.
Extensive experiments show the effectiveness of the proposed method, surpassing previous SOTA by a significant margin.

**Strengths:**

+ This paper is well-written in English. The presentation is clear and very easy to follow. All necessary technical details for reproducing the method are provided in the paper and supplementary material.
+ The method is valid and sound. Learning the spatial relationships is essential for 3D visual grounding, and it was handled well by the proposed network.
+ The proposed method achieves SOTA results on established benchmarks, showing great improvements over previous methods.

**Weaknesses:**

I didn't find any significant technical flaws in the paper. But I do have some concerns that prevent me from giving a higher recommendation.

1) The inputs to the relation matching module seem highly engineered, as detailed in the supplementary material. Although the auxiliary features are shown to be effective for the final visual grounding accuracy, I wonder if the authors can justify the impact brought by those features vs the proposed network itself. That is, if the same set of features was fed to a similar architecture, e.g. BUTD-DETR, would it achieve better performance than before?
2) The object detector receives language inputs. How does it perform against other transformer-based 3D detector, e.g. 3DETR? If the language inputs were removed from the detector but fed into the later modules, how would the performance change?
3) Regarding the experiments on ScanRefer, can the authors provide results on the online benchmark? The performance on the test set is also critical for validating the claim of achieving the SOTA performance.
4) Can the authors provide more qualitative results on ScanRefer?

**Questions:**

Please see the Weaknesses section.

**Limitations:**

The limitations are properly discussed.

---

> ### Author Rebuttal · Authors · 2023-08-10
>
> We thank the reviewer for the detailed review and the positive feedback on our paper. We appreciate your insights and concerns, and below are our responses to the points you raised.
>
> **Q1:  The impact of designed spatial relation features (inputs to the relation matching module).**
>
> **A1:**
> The spatial relation features are devised to directly represent the commonly described relationships between objects. It is worth noting that these relation features are closely tied to our approach. We calculate these features for the objects detected by the prior text-guided detection network. If the contextual objects are not explicitly detected, it will be difficult to compute these relation features and demonstrate their full effectiveness.
>
> To evaluate the isolated impact brought by these spatial features, we incorporate them into the BUTD-DETR. The table below illustrates that incorporating these relation features improves performance in BUTD-DETR. However, this improvement is less significant compared to what our method achieves. Our method learns to detect the contextual objects along with the target, allowing for better utilization of their spatial relation features for target grounding.
> | Method          |  Acc@0.25 |
> |----------------------|:---------:|
> | BUTD-DETR            |   43.3   |
> | BUTD-DETR (+relation feats.) | 45.59 |
>
> **Q2: Performance of the text-guided object detector.**
>
> **A2:** We compare the performance of our model's detection network with 3DETR on the ScanNet dataset. As shown in the table below, our detection network does not perform as well as 3DETR.  This could be because we employ a compact detection network with only 3 transformer decoder layers, which may not be advantageous for general object detection. To enhance the final grounding performance, it may be beneficial to focus on improving the performance of the detection network in the model.
>
> | Method          |  mAP@0.25 |
> |----------------------|:---------:|
> | 3DETR            |   61.1 |
> | The detection network in our model| 57.3|
>
> For our method, we remove the language inputs from the detection network and evaluate its grounding performance on the Nr3D dataset. Removing the language inputs leads to significant performance degradation. Without the description information in the first detection stage, the model may not be able to detect related objects, thus limiting the follow-up grounding performance. This illustrates the importance of accurate object detection in the first stage.
> | Method          |  Acc@0.25 |
> |----------------------|:---------:|
> | CORE-3DVG            |   49.57 |
> | CORE-3DVG (w/o language for detection)| 20.68|
>
> **Q3: Evaluation on online ScanRefer benchmark.**
>
> **A3:**
> Thank you for your advice. We have submitted our method to the online ScanRefer benchmark. **Our method, CORE-3DVG, achieves **the best performance** in overall Acc\@0.25 (60.11%), surpassing the second-ranked method by a significant margin of 5.78 points.** Our performance in overall Acc\@0.5 is nearly on par with the current leader, achieving 45.27% compared to their 45.45%. It's worth noting that our method employs a compact detection network with only 3 decoder layers, which may limit its advantages in evaluations requiring more precise localization. Nevertheless, we still achieve the best performance in both Acc\@0.25 and Acc\@0.5 on the more challenging "Multiple" subset. Our method also outperforms all published methods across all settings. These results further validate our claim of SOTA performance.
>
> The table below compares our method with leading published methods. For more detailed comparisons, please visit the official ScanRefer benchmark website. We will update the paper with these results.
> | Method           |   Unique  |   Unique  |  Multiple |  Multiple |  Overall  |  Overall  |
> |------------------|:---------:|:---------:|:---------:|:---------:|:---------:|:---------:|
> |                  |  Acc@0.25 |  Acc@0.5  |  Acc@0.25 |  Acc@0.5  |  Acc@0.25 |  Acc@0.5  |
> | ScanRefer        |    68.6   |    43.5   |    34.9   |    21.0   |    42.4   |    26.0   |
> | TGNN             |    68.3   |    58.9   |    33.1   |    25.3   |    41.0   |    32.8   |
> | InstanceRefer    |    77.8   |    66.7   |    34.6   |    26.9   |    44.3   |    35.8   |
> | D3Net            |    79.2   |    68.4   |    39.1   |    30.7   |    48.1   |    39.2   |
> | HAM              |    78.0   |    63.7   |    41.5   |    33.2   |    49.7   |    40.0   |
> | 3DVG-Transformer |    77.3   |    57.9   |    43.7   |    31.0   |    51.2   |    37.0   |
> | 3DJCG            |    76.8   |    60.6   |    43.9   |    31.2   |    51.3   |    37.8   |
> |**CORE-3DVG (ours)**| **85.57** | **68.67** | **52.75** | **38.50** | **60.11** | **45.27** |
>
>
> **Q4: More qualitative results on ScanRefer.**
>
> **A4:**
> We have included qualitative examples on ScanRefer in the attached PDF of the global author rebuttal. These examples help to provide a better understanding of how our method performs in various scenarios. Additionally, we compare our method with BUTD-DETR to highlight the advantages of leveraging contextual objects and relations for target localization.

---

> > ### Comment · Reviewer_oYfR · 2023-08-11
> >
> > I would like to thank the authors for composing such a detailed response. All my concerns were properly addressed. In addition, the SOTA performance (especially the ones on the "Multiple" split) on the benchmark looks convincing to me. To re-iterate: The proposed spatial relationship modelling via pseudo-label generation is a critical contribution to this field, where methods to overcome 3D vision-language data scarcity are desperately needed. Therefore, I will keep my original rating, and vote for Acceptance.

---

> > > ### Author Response · Authors · 2023-08-20
> > >
> > > Dear Reviewer,
> > >
> > > Thank you for your thoughtful comments and for pointing out the significance of our work.
> > >
> > > We are glad to hear that our response has successfully addressed your concerns. Your acknowledgment of the significance of our spatial relationship modeling and pseudo-label generation in addressing the challenges of 3D vision-language data scarcity is greatly appreciated.  **By the way**, in light of your earlier comments about certain concerns preventing a higher recommendation, we would like to know if there are any remaining issues. We would appreciate your suggestions and will make improvements based on them. 🤗
> > >
> > > Once again, thank you for your support and feedback. Your suggestions have been very helpful in improving our work and its presentation. We look forward to contributing further to the community.
> > >
> > > Best regards!

---

### Official Review · Reviewer_2pN3 · 2023-07-05

**Soundness:** 2 fair
**Presentation:** 2 fair
**Contribution:** 2 fair
**Rating:** 4
**Confidence:** 5

**Summary:**

This paper introduces the CORE-3DVG method, which consists of three consecutive modules: Text-guided object detection network for detecting all candidates, Relation matching network for matching the relationships among candidate objects, and Target inference network for selecting the referred object. The proposed method incorporates a self-generation strategy to avoid the need for additional supervision. As a result, the CORE-3DVG method achieves state-of-the-art (SOTA) performance across multiple datasets.

**Strengths:**

1. The underlying idea of CORE-3DVG involves the explicit extraction of candidate objects and the reliable modeling of their relationships. The proposed module for extracting nouns/weakly supervised relational information appears to be a plausible and effective solution to avoid the need for additional data. By incorporating this approach, the CORE-3DVG method achieves a reliable and guaranteed modeling of candidate object relationships.

2. The improvement achieved by CORE-3DVG on multiple general datasets is significant, as supported by additional ablation experiments that demonstrate the effectiveness of each module.

3. The overall coherence and comprehensibility of the paper make it easy to follow.

**Weaknesses:**

1. The paper lacks essential analysis regarding the pseudo labels.
    * For methods like CORE-3DVG, which rely on pseudo labels to avoid additional supervision, the quality of the pseudo labels is undeniably crucial. While the appendix provides detailed information about the generation process and the main text explains how the pseudo labels are generated, and the experiments indirectly demonstrate the usefulness of the pseudo labels, it is necessary to conduct a thorough analysis of the pseudo labels.
    * For example, it would be valuable to assess the accuracy of localizing objects other than the target noun under the given annotation with only target nouns.
    * Additionally, it is important to evaluate whether the relationships among candidate objects are correctly modeled under weak supervision.

2. Lack of detailed comparisons and discussions in the paper. Most of the methods listed in Table 1 claim to model the relationships between the target and surrounding objects. For instance, even Instancerefer includes an explicit relationship modeling module and scores. I am curious whether the significant performance improvement is indeed attributed to the explicit pseudo labels (which makes the analysis in point 1 particularly important). If so, the authors should enhance the discussion in this regard to further establish the positioning of CORE-3DVG among a range of 3DVG papers. A thorough discussion of the similarities and differences with previous methods would not diminish the contributions of this paper.

**Questions:**

Pls refer to weaknesses.

**Limitations:**

This paper discusses the limitations of generalization ability.

---

> ### Author Rebuttal · Authors · 2023-08-09
>
> We appreciate the constructive suggestions and comments. The following are our responses to the comments.
>
> **Q1: The lack of essential analysis regarding pseudo labels.**
>
> **(a) The accuracy of localizing objects other than the target.**
>
> **A1 (a):** Thanks for your suggestions. For a language description, correctly localizing mentioned contextual objects is indeed important for pinpointing the target object. However, the datasets (Nr3D, ScanRefer) with natural (human-annotated) descriptions lack the labels for mentioned contextual objects, making it infeasible to directly assess the accuracy of localizing such unlabeled objects other than the target. In light of this, we use the Sr3D dataset as a benchmark for this evaluation. Sr3D is built upon the ScanNet dataset and synthesizes template-based descriptions that define a relationship between a target and a single surrounding object (from ScanNet objects). Sr3D has the labels of the target object and its surrounding object.
>
> To evaluate the ability to detect objects other than the target, we take the model trained on the Nr3D dataset (based on pseudo labels) and evaluate its accuracy in localizing surrounding objects on the Sr3D test set. Given that our detector produces detection results for all mentioned objects for input descriptions, we take these detection results to compute the recall and mean IoU for localizing the surrounding objects. The evaluation results are listed in the table below. Although our model is not directly trained on the Sr3D dataset, it is able to detect surrounding objects with high recall and mean IoU. This demonstrates the effectiveness of pseudo labels for learning objects other than the target.
> |                                | Recall (IoU>0.25) | Mean IoU |
> |--------------------------------|:------:|:--------:|
> | Object Detections (all)        |  85.84 |   51.77  |
> | Object Detections (score>0.25) |  74.75 |   45.89  |
> | Object Detections (score>0.5)  |  71.29 |   43.92  |
>
> Moreover, for the dataset subsets where surrounding objects are successfully detected or not (denoted by "Context Detected" and "No Context Detected", respectively), we compute the accuracy of 3D visual grounding separately. As shown in the table below, the performance is significantly higher when surrounding objects are successfully detected. This illustrates the importance of detecting contextual objects other than the target object for 3D visual grounding.
> |  Dataset Subsets    | Acc\@0.25               |
> |---------------------|:-----------------------:|
> | Context Detected    |          50.31          |
> | No Context Detected |          35.46          |
>
> **(b) Evaluate whether the relationships among candidate objects are correctly modeled under weak supervision.**
>
> **A1 (b):**
> Similarly, we use the model trained on the Nr3D dataset and evaluate its accuracy of relation modeling on the Sr3D dataset.  In our method, we perform weak supervision on the matching scores of the modeled object relations, with the aim of helping the network distinguish the target from other objects. To evaluate these modeled relations, we analyze their matching scores for identifying the target.
> Specifically, we calculate the relation matching scores between all detected objects and the detected surrounding object. Then, we compute the top-K accuracy of target identification (which indicates whether the target object is among the top-K matched objects of the surrounding object). From the table below, we can see the modeled relations alone are effective in recognizing targets from other objects.
> |  Top-1 Acc |  Top-2 Acc |  Top-4 Acc|  Top-6 Acc|  Top-8 Acc |
> |:-----:|:-----:|:-----:|:-----:|:-----:|
> | 40.0% | 53.9% | 69.7% | 78.6% | 84.9% |
>
> We further evaluate the final grounding accuracy on dataset subsets where the target object is matched by the top-K scoring relations of the detected surrounding objects (denoted by "Top-K Subset"). As illustrated in the table below, when the target is more accurately associated with  the modeled relations, the model also exhibits higher accuracy in 3D visual grounding. These results further demonstrate the importance of accurately modeling contextual relations for final target identification.
> |              | Acc\@0.25 |
> |--------------|:--------:|
> | Top-1 Subset |   81.2   |
> | Top-2 Subset |   76.9   |
> | Top-4 Subset |   72.2   |
> | Top-6 Subset |   68.9   |
> | Top-8 Subset |   66.6   |
>
> **Q2: Lack of detailed methodological comparisons and discussions in the paper.**
>
> **A2:** Thanks for your advice. We will add a more detailed comparison and discussion in the revision.
> Most previous works typically match a large set of pre-prepared object proposals with the language description for target identification, without localizing the mentioned contextual objects related to the target, which is important as we analyzed above. These previous works largely rely on graph networks and transformer architectures to model the relationships between objects. For example, InstanceRefer considers k nearest neighbors for each candidate object to construct the object graph and perform feature fusion from local neighborhoods, which could be less flexible to infer relations such as "farthest from" in a language description.
>
> In contrast, our method proposes to explicitly model contextual objects and relations, enabling a more reliable target reasoning. (1) The explicit pseudo labels help our network find the semantically related objects for a given description, allowing us to focus on these objects to capture contextual information.
> (2) Based on (1), we devise various spatial relation features for detected objects and perform weak supervision to model important contextual relations, which helps distinguish the target object.
> Our analysis above and the ablation studies in the paper illustrate the importance and efficacy of the proposed methods. We will add the above results and analyses to the revision.

---

> ### Author Response · Authors · 2023-08-20
> **Looking forward to your comments.**
>
> Dear Reviewer,
>
> Thank you for your thorough review and constructive feedback on our method. We have taken your comments seriously and crafted a detailed rebuttal. Here's a concise summary:
> 1. **Pseudo labels analysis:**
>   - **Localization of non-target objects:** Due to the lack of labels for contextual objects in Nr3D and ScanRefer datasets, we utilized the Sr3D dataset for evaluation. Our model, trained on Nr3D, demonstrated high recall and mean IoU for detecting surrounding objects in Sr3D, emphasizing the effectiveness of pseudo labels. Furthermore, our results show a significant performance boost in 3D visual grounding when surrounding objects are detected.
>   - **Modeling relationships under weak supervision:** Using the Sr3D dataset, we evaluated our method's ability to model relationships. The results indicate that our modeled relations are effective in distinguishing targets from other objects. The grounding accuracy also improves when the target is more accurately associated with the modeled relations.
> 2. **Methodological comparisons:**
>   - We acknowledge the need for a more detailed comparison and we will enhance our discussion in the revision. Unlike most previous works that match pre-prepared object proposals with language descriptions, our method emphasizes explicitly modeling contextual objects and relations. This approach, bolstered by our pseudo labels, allows for more reliable target reasoning. Our analyses and ablation studies further highlight the importance and efficacy of our method.
>
> We hope that our responses address your concerns and we would be grateful if you could consider raising your score in light of our explanations. We are committed to improving our paper and would greatly appreciate any further feedback or suggestions.
>
> Thank you once again for your time and consideration.
>
> Best regards!

---

### Official Review · Reviewer_DS3o · 2023-07-06

**Soundness:** 3 good
**Presentation:** 3 good
**Contribution:** 3 good
**Rating:** 6
**Confidence:** 4

**Summary:**

This work presents a novel approach for learning 3D visual grounding: the task of localizing and identifying objects given natural language inputs. The language prompts in this problem often involve using other contextual objects and their relations to the target object. The proposed approach breaks down this problem into an intuitive set of transformer-based architecture components:
- a module to localize the objects corresponding to the nouns in the prompt and produce object features
- a module to model the spatial relations between the detected objects conditioned on the text prompt
- a module to combine the object features and learned spatial relations with the text prompt to predict the target object

Prior work has generally been evaluated on ground truth 3D bounding boxes and on 3D detector outputs. This work takes the practically feasible approach of evaluating prior work on 3D detector outputs, and in this case the proposed work outperforms prior methods on three datasets by a significant margin. Extensive and detailed ablation studies justify the design of each of the components of the architecture.

**Strengths:**

Originality
- Most prior work does not treat 3D visual grounding as an end-to-end learning problem, relying on pre-trained detectors for 3D bounding box proposals, whereas the proposed approach does.
- The most successful prior work that tries to use transformers for this task, BUTD-DETR, uses a transformer encoder-decoder setup like DETR that implicitly combines the 3D bounding box, visual feature and text feature information with cross-attention. The proposed work breaks the visual grounding problem using multiple transformer-based modules, into text-guided object detection, modeling spatial relations between objects, and target object inference based on the output of the other two modules. This is empirically shown to be more superior to BUTD-DETR.
- Explicitly detecting objects based on a text prompt is challenging because the labels for visual grounding only contain annotations for the target object and not the context objects. This work proposes an effective pseudo-labelling technique to address this.

Quality
- The proposed work has significant performance gains over prior methods, verifying the effectiveness of its design.
- The qualitative results demonstrate high performance, including cases with complex text prompts describing context/target object relations and small objects.

Clarity - The paper is well written but quite dense due to having to contain descriptions for all the separate modules and their design.

**Weaknesses:**

Clarity
- The draft appears to have been compiled with an altered value for the general line spacing value for the text paragraphs. The text appears vertically more tight than the standard NeurIPS. This is against the submission guidelines and should be fixed.
- Section 3.2 about pseudo-label generation was a bit hard to follow, it would be beneficial to revise it for clarity or include a visual explanation using a diagram in the supplement.

Quality
- While in aggregate the performance is better than prior work, the current draft could benefit from more detailed quantitative comparisons with prior work beyond the results in Table 2 for unique vs multiple objects. How can we further characterize the examples where the proposed approach performs better than prior work e.g. small objects or cluttered scenes? Why exactly does this representation outperform prior work?
- This similarly follows for qualitative examples as well, it would be beneficial to visually compare and understand the advantages of the proposed approach with prior work.

**Questions:**

- What are more detailed reasons for why exactly the proposed approach which breaks down the problem into smaller sub-problems works better than prior work? This is valuable insight that would strengthen the paper
- Is it possible to show qualitative examples comparing the proposed approach with prior methods?

**Limitations:**

The authors have adequately addressed the limitations and potential negative societal impact.

---

> ### Author Rebuttal · Authors · 2023-08-10
>
> We thank the reviewer for the positive feedback and insightful comments. Below, we address each of the concerns raised.
>
> **Clarity**
>
> **Response:**
> Thanks for the kind reminder. We will revise the paper and move some details to the supplementary material to make it more concise and clear. The text in Section 3.2 will be improved, and we will include a visual diagram in the supplementary material to provide a clearer understanding of the pseudo-label generation process.
>
> **Quality**
>
> **Q1: More detailed quantitative comparisons with prior work beyond the results in Table 2.**
>
> **A1:** Thanks for the suggestions.
> To provide a more comprehensive comparison between our method and the prior work BUTD-DETR, we evaluate them on the ScanRefer dataset from various aspects, which includes the number of mentioned objects, the object size, and the total number of objects in each scene).
>
> - **The number of mentioned objects**: We first extract the noun phrases of mentioned objects from the language descriptions, and divide the dataset into subsets based on the number of noun phrases. We evaluate the performance of our method and BUTD-DETR on each subset, as shown in the table below. It can be seen that our method performs similarly to BUTD-DETR when only one object is mentioned, but significantly outperforms it when two or more objects are mentioned. This suggests that our method excels at utilizing contextual object information for target grounding, as descriptions with multiple nouns typically describe the target along with its associated contextual objects.
> | |  #Nouns=1 |  #Nouns=2 |  #Nouns=3 | #Nouns>=4 |
> |----------------------|:---------:|:---------:|:---------:|:---------:|
> | BUTD-DETR            | **50.42** |   53.16   |   51.15   |   51.09   |
> | **CORE-3DVG (ours)** |   50.14   | **56.78** | **58.68** | **56.51** |
>
> - **The object size**: We also divide the dataset into subsets based on the sizes of target objects and compare our method with BUTD-DETR separately on these subsets. The table below shows that our method brings larger improvements over BUTD-DETR in localizing smaller targets than larger ones. Identifying small targets solely based on their appearances is more challenging, but our explicit modeling of contextual objects and relations improves the identification of small targets.
> | Object size          |  ≤$0.4^3$ | $[0.4^3\sim 0.8^3]$ |  >$0.8^3$ |
> |----------------------|:---------:|:-------------------:|:---------:|
> | BUTD-DETR            |   45.79   |        47.53        |   65.34   |
> | **CORE-3DVG (ours)** | **51.09** |      **53.24**      | **68.70** |
>
> - **The total number of objects in each scene**: We use the total number of objects in each scene to roughly indicate the level of clutter and compare our method with BUTD-DETR on subsets with different ranges of object numbers. From the results listed in the table below, our model achieves higher improvements on subsets containing 30\~40 and 40\~50 objects, particularly a significant 7.84 point improvement on the subset of 40\~50 objects. This demonstrates our advantages in identifying the target when lots of objects are present in the scene. Our method is developed to first detect semantically mentioned objects and focus on these objects to capture contextual information, which can reduce distractions from unrelated objects and better handle cluttered scenes.
> |#Objects|    ≤20    |   20\~30   |   30\~40   |   40\~50   |    >50    |
> |----------------------|:---------:|:---------:|:---------:|:---------:|:---------:|
> | BUTD-DETR            |   57.30   |   57.56   |   50.39   |   49.08   |   45.60   |
> | **CORE-3DVG (ours)** | **61.26** | **62.05** | **55.67** | **56.93** | **49.28** |
>
> **Q2: Visually compare the proposed approach with prior work.**
>
> **A2:** Thanks for your advice. We have submitted a PDF in the global author rebuttal, which presents the visual grounding results of our method compared to the ground truth and the results of BUTD-DETR on both the Nr3D and ScanRefer datasets.  Overall, our method demonstrates a better understanding of the contextual description about the target, and is able to pinpoint the target with the help of the modeled contextual object and relations.
>
> **Questions**
>
> **Q1: Reasons for breaking down the problem.**
>
> **A1:**
> Our idea of decomposing the 3D grounding problem into sub-problems is motivated by the human cognitive process of localizing visual targets based on language descriptions. When we humans interpret a textual description to identify a target object in a scene, we first need to identify the objects mentioned in the text, and then determine the correct target in relation to its contextual objects based on the described object relations.
>
> By breaking down the 3D grounding into sub-problems, we can also explicitly model the contextual objects and relations associated with the target object. For free-form language descriptions, there are no explicit labels for objects and their relations outside the referred target. This necessitates specialized modeling methods for learning both the detection of contextual objects and the understanding of their relations with the target. Hence, our approach models these two sub-problems separately.  Compared to prior works, our explicit modeling of contextual objects and relations allows us to distinguish targets more intuitively and accurately, resulting in better performance and interpretability. We will revise the paper to elucidate the advantages of our approach over previous work more clearly.
>
>
> **Q2:Qualitative examples comparing the proposed approach with prior methods.**
>
> **A2:**
> We have included qualitative examples in the attached PDF in the global author rebuttal. These examples showcase the performance of our method in comparison to the ground truth and the prior method BUTD-DETR, highlighting the advantages of our method in exploiting contextual objects and relations for target reasoning.

---

> > ### Comment · Reviewer_DS3o · 2023-08-21
> > **Thank you for the detailed response**
> >
> > I would like to thank the authors for this excellent response. The stratified quantitative results clearly show the strengths of the proposed approach. The revision describing the motivation behind the model design will be a great addition to the paper, as well as the qualitative examples. At this point I will increase my rating to WA.

---

> > > ### Author Response · Authors · 2023-08-21
> > >
> > > Dear Reviewer,
> > >
> > > Thank you for your positive feedback and support. We are glad to hear that our response was well-received. Your suggestions have been valuable in helping us to better showcase the advantages of our method and have undoubtedly improved the quality of our paper. We will incorporate the discussed results and improvements into the revised version.
> > >
> > > Best regards!

---

> ### Author Response · Authors · 2023-08-20
> **Looking forward to your comments.**
>
> Dear Reviewer,
>
> Thank you for your insightful feedback on our submission. We appreciate the time and effort you've invested in reviewing our work. In response to the points you raised, we've provided a concise summary of our rebuttal:
> 1. **Clarity:** We are committed to revising the paper for better clarity, especially Section 3.2, and we will include a visual diagram in the supplementary material to aid understanding.
> 2. **Quality:**
>   - **Quantitative Comparisons:** We have conducted additional evaluations on the ScanRefer dataset, considering various aspects like the number of mentioned objects, object size, and scene clutter. Our method consistently outperforms BUTD-DETR, especially in scenarios with multiple mentioned objects, smaller target objects, and cluttered scenes.
>   - **Visual Comparisons:** We have attached a PDF in the global author rebuttal, which visually contrasts our method with BUTD-DETR, demonstrating our method's superior understanding of contextual descriptions.
> 3. **Questions:**
>   - **Decomposition Rationale:** Our approach of decomposing the 3D grounding problem is inspired by human cognition.  We believe that by modeling the contextual objects and relations separately, we can achieve more intuitive and accurate target identification. This is reflected in our improved performance and interpretability.
>   - **Qualitative Examples:** The attached PDF in the global author rebuttal provides qualitative examples, emphasizing our method's advantages in leveraging contextual objects and relations.
>
> We hope that our responses address your concerns and we would be grateful if you could consider raising your score in light of our explanations. We are committed to improving our paper and would greatly appreciate any further feedback or suggestions.
>
> Thank you once again for your time and consideration.
>
> Best regards!

---

### Official Review · Reviewer_Dhe4 · 2023-07-07

**Soundness:** 2 fair
**Presentation:** 2 fair
**Contribution:** 2 fair
**Rating:** 4
**Confidence:** 3

**Summary:**

This paper proposes to explicitly model object relations mentioned in the language description for 3D visual grounding task. The proposed method designs a relation matching network and weakly-supervised contextual relation learning strategy to perform contextual learning. Finally, they design a target inference network to output the referred object and give performance of model on three benchmarks.



**Strengths:**

1.	This paper proposes to explicitly model the relation mentioned in the context for 3D visual grounding.
2.	The paper introduces pseudo-label self-generation method and weakly-supervised contextual relation learning to address the unlabeled objects.
3.	Extensive ablation experiments demonstrate the effectiveness of the proposed modules.


**Weaknesses:**

1.	The comparison of experimental results misses the ViL3DRel method and 3D-SPS[17] method which use ground truth of all proposals in a scene for training. Maybe you can use ground truth of all proposals in a scene for a fair comparison.
2.	It seems that the paper uses the b_gt to supervise the detection results of labeled noun phrase. Why not supervise the detection results of the language description?
3.	The overall optimization objectives of contextual relation learning is very unreasonable in line 210. For a target object, is there only one relationship possible?
4.	Why does the detection network have the ability to detect the unlabeled objects without any other annotations in section 3.2? Maybe you can explain it more clearly.


**Questions:**

See weakness.

---

> ### Author Rebuttal · Authors · 2023-08-09
>
> We thank the reviewer for the valuable suggestions and comments. Below are our responses to the raised questions.
>
> **Q1: Comparison with methods using ground truth object proposals.**
>
> **A1:** In our approach, we focus on reasoning about the target objects from point clouds without relying on pre-provided object proposals. Consequently, we did not compare with methods that utilize ground truth proposals in our original manuscript.
> But your suggestion can help us better understand the effectiveness of our proposed modules. Following your suggestion, we compare our method with ViL3DRel[27], 3D-SPS[17], and the Multi-view [13] method. For a fair comparison, as in previous works, we use PointNet++ to encode point cloud features of ground truth object proposals and take them as inputs to train our model. As shown in the following table, our approach consistently outperforms previous leading methods across various settings. Even compared with the ViL3DRel (w/ KD) that has applied knowledge distillation based on a powerful teaching model, our method still achieves better performance. We will add more detailed results in the revised paper.
>
> | Method            |  Overall  |    Easy   |    Hard   | View-dep. | View-indep. |
> |-------------------|:---------:|:---------:|:---------:|:---------:|:-----------:|
> | 3D-SPS            |    51.5   |    58.1   |    45.1   |    48.0   |     53.2    |
> | Multi-view        |    55.1   |    61.3   |    49.1   |    54.3   |     55.4    |
> | ViL3DRel (w/o KD) |    58.1   |     -     |     -     |     -     |      -      |
> | ViL3DRel (w/ KD)  |    64.4   |    70.2   |    57.4   |    62.0   |     64.5    |
> | **CORE-3DVG (ours)**  | **66.60** | **72.64** | **60.80** | **65.79** |  **66.90**  |
>
>
> **Q2: Why not use the $b_{\rm gt}$ to supervise the detection results of the language description?**
>
> **A2:** Sorry for the ambiguity. The target object label $b_{\rm gt}$ is indeed employed to supervise the detection results of the entire language description (as shown in Eq. 5). We develop our model to first detect all semantically mentioned objects (including target and contextual objects) and subsequently identify the target object by analyzing inter-object relations. The target object label $b_{\rm gt}$ plays a pivotal role in model training in the first detection stage. We will clarify this during revision.
>
> **Q3: The question about the overall optimization objectives of contextual relation learning in line 210. Does it mean that there is only one relationship for a target object?**
>
> **A3:** No, it does not mean that there's only one relationship for a target object but **at least** one relationship. We will modify the statement in line 210 as "For the target object, there exists at least one contextual relation with a high matching score."
> Specifically, given the absence of annotations for contextual objects and their relations, we employ weak supervision to exploit important contextual information. The objective 1 aims to learn the contextual object that is most related to the target object. For the most related contextual object, the objective 2 ensures that its relation to the target object is distinct from its relations to others, helping to identify the target object. We will improve the clarity of this in the revised manuscript.
>
>
> **Q4: Explain more clearly in Section 3.2 why the detection network has the ability to detect unlabeled objects without any other annotations?**
>
> **A4:** Sorry for the lack of clarity here. We propose to generate pseudo labels for unlabeled objects, enabling the network to learn to detect them. Specifically, for the entire language descriptions, we only have labels for the referred target objects, but lack labels for other objects that are also mentioned in the descriptions. This makes it difficult for the network to learn to detect all the mentioned objects, which limits the follow-up grounding performance.
>
> To circumvent this, we extract the object phrases from the language descriptions, thus obtaining a diverse set of phrases corresponding to labeled target objects and unlabeled contextual objects. Using the phrases and labels of the target objects, we train the network to detect objects based on input noun phrases. As a result, the network is also able to detect the unlabeled contextual objects based on their noun phrases. These detection results are utilized as pseudo labels, which facilitate the learning process to detect all mentioned objects (both target and unlabeled objects) based on the complete language description.

---

> ### Author Response · Authors · 2023-08-20
> **Looking forward to your comments.**
>
> Dear Reviewer,
>
> Thank you again for your valuable feedback on our submission. We appreciate the time and effort you've dedicated to reviewing our work. Here's a summary of our responses:
> 1. **Comparison with methods using ground truth object proposals**: Upon your suggestion, we have now included comparisons with ViL3DRel [27], 3D-SPS [17], and the Multi-view [13] method, showing that our method consistently outperforms these methods across various settings.
> 2. **Supervision of detection results**: We apologize for any ambiguity. To clarify, the target object label $b_{\rm gt}$ is indeed used to supervise the detection results of the entire language description. This will be clarified in the revised version.
> 3. **Optimization objectives of contextual relation learning**: We have clarified our statement in line 210 to emphasize that a target object has **at least** one relationship, not just one. We employ weak supervision to exploit important contextual information, and this will be elaborated upon in the revision.
> 4. **Detection of unlabeled objects in Section 3.2**: We generate pseudo labels for unlabeled objects, enabling the network to detect them. This process will be explained more clearly in the revised manuscript.
>
> We hope that our responses address your concerns and we would be grateful if you could consider raising your score in light of our explanations. We are committed to improving our paper and would greatly appreciate any further feedback or suggestions.
>
> Thank you once again for your time and consideration.
>
> Best regards!

---

### Author Rebuttal · Authors · 2023-08-10

We sincerely thank all reviewers for their constructive feedback and valuable insights. We have carefully addressed each concern raised and will incorporate the suggested changes to improve the clarity and quality of our paper.

In response to the concerns raised, we have made more comprehensive comparisons to demonstrate the superiority of our method. Additionally, we have provided detailed explanations of our methodology. We have also enhanced our analysis regarding pseudo labels, and conducted a thorough evaluation of our method's performance under different scenarios. Furthermore, we submitted our method to the online ScanRefer benchmark, where it achieved state-of-the-art performance, further validating our claims.

We believe that these revisions will significantly enhance the paper's clarity and its potential value to the community. We appreciate the opportunity to clarify our contributions and thank the reviewers for their time and effort in reviewing our work.

---

> ### Author Response · Authors · 2023-08-21
>
> Dear Area Chair and Reviewers,
>
> Thank you for the time and effort spent reviewing our paper. We have thoroughly considered all the comments to improve our work.
>
> We are thankful to reviewers DS3o (R2), oYfR (R4), and VN72 (R5) for their positive feedback and for recognizing the potential of our research. We have also diligently addressed the issues raised by reviewers Dhe4 (R1), 2pN3 (R3), and we hope our clarifications and additional analyses have resolved their concerns.
>
> The constructive suggestions from all reviewers have undoubtedly enriched our paper, and we are truly appreciative of this collaborative review process. Once again, thank you all for your dedication!
>
> Warm regards!

---

### Decision · Program_Chairs · 2023-09-21

**Decision:**

Accept (poster)

**Comment:**

The paper proposes a new model for 3D visual grounding which explicitly learns to detect context objects and relations. Reviewers appreciated the novelty of the idea, good experimental results over state of the arts, and thorough ablations provided in the rebuttal. Three out of five reviewers gave the weak accept scores after the rebuttal. The remaining two reviewers did not reply to the authors, but the AC considered the authors have adequately addressed their concerns in the rebuttal including the in-depth analysis on the quantitative results and pseudo labels. Therefore, the AC recommends accepting the paper. The authors should revise the paper according to reviewers’ comments in their final version.